

# Year-round CH$_4$ and CO$_2$ flux dynamics in two contrasting freshwater ecosystems of the subarctic

Mathilde Jammet[1], Sigrid Dengel[2], Ernesto Kettner[1], Frans-Jan W. Parmentier[3], Martin Wik[4], Patrick Crill[4], Thomas Friborg[1]

[1]Center for Permafrost (CENPERM), Department for Geosciences and Natural Resource Management, University of Copenhagen, Copenhagen, 1350, Denmark

[2]Climate and Ecosystem Sciences Division, Lawrence Berkeley National Laboratory, Berkeley CA, 94720, USA

[3]Department of Soil Quality and Climate Change, Norwegian Institute of Bioeconomy Research (Nibio), Høyskoleveien 7, 1430 Ås, Norway

[4]Department of Geological Sciences, Stockholm University, Stockholm, SE-106 91, Sweden

*Correspondence to*: Mathilde Jammet (mathilde.jammet@ign.ku.dk)



**Abstract.** Lakes and wetlands, common ecosystems of the high northern latitudes, exchange large amounts of the climate-forcing gases methane ($CH_4$) and carbon dioxide ($CO_2$) with the atmosphere. The magnitude of these fluxes and the processes driving them are still uncertain, particularly for subarctic and Arctic lakes where direct measurements of $CH_4$ and $CO_2$ emissions are often of low temporal resolution and are rarely sustained throughout the entire year.

Using the eddy covariance method, we measured surface-atmosphere exchange of $CH_4$ and $CO_2$ during 2.5 years in a thawed fen and a shallow lake of a subarctic peatland complex. Gas exchange at the fen exhibited the expected seasonality of a subarctic wetland with maximum $CH_4$ emissions and $CO_2$ uptake in summer, as well as low but continuous emissions of $CH_4$ and $CO_2$ throughout the snow-covered winter. The seasonality of lake fluxes differed, with maximum $CO_2$ and $CH_4$ flux rates recorded at spring thaw. During the ice-free seasons, we could identify surface $CH_4$ emissions as mostly ebullition

events with a seasonal trend in the magnitude of the release, while a net $CO_2$ flux indicated photosynthetic activity. We found correlations between surface $CH_4$ emissions and surface sediment temperature, as well as between diel $CO_2$ uptake and diel solar input. At ice-out, the breakdown of thermal stratification following ice thaw triggered the degassing of both $CH_4$ and $CO_2$. This spring burst was observed in two consecutive years for both gases, with a large inter-annual variability in the magnitude of the $CH_4$ degassing.

On the annual scale, spring emissions converted the lake from a small $CO_2$ sink to a $CO_2$ source. 80 % of total annual carbon emissions from the lake were emitted as $CO_2$. The annual total carbon exchange per unit area was highest at the fen, which was an annual sink of carbon with respect to the atmosphere. Continuous respiration during the winter partly counteracted the fen summer sink by accounting for, as both $CH_4$ and $CO_2$, 33 % of annual carbon exchange. Our study underlines (1) the importance of overturn periods (spring or fall) for the annual $CH_4$ and $CO_2$ emissions of northern lakes, (2) the significance

of lakes as atmospheric carbon sources in subarctic landscapes while fens can be strong carbon sink and (3) the potential for ecosystem-scale eddy covariance measurements to improve the understanding of short-term processes driving lake-atmosphere exchange of $CH_4$ and $CO_2$.

**Keywords**: eddy covariance, methane, carbon dioxide, lake, wetland, artificial neural networks, carbon budget, subarctic



## 1. Introduction

Lakes and wetlands are linked to the atmospheric carbon pool via the exchange of methane ($CH_4$) and carbon dioxide ($CO_2$), which are two important climate-forcing gases (Myhre et al., 2013). While wetlands have been a focus of study due to their high $CH_4$ source function (Christensen et al., 2003; Crill et al., 1988; Olefeldt et al., 2013) and carbon sequestration

capacity (Kayranli et al., 2009; Whiting and Chanton, 2001), lakes have only recently been incorporated together with streams as a separate source in global $CH_4$ budgets with an uncertain global emission rate of 8–73 Tg $CH_4$ $yr^{-1}$ (Kirschke et al., 2013). The low number of experimental studies and the variability in the magnitude of emissions across lake types (Wik et al., 2016) explains some of this order of magnitude scale uncertainty. Carbon emissions from lakes outweigh part of the land carbon sink, because they emit $CH_4$ (Bastviken et al 2011) and because they respire as $CO_2$ a portion of the carbon that

is transported laterally from terrestrial soils to lakes (Algesten et al., 2004; Battin et al., 2009; Cole et al., 2007; Tranvik et al., 2009). Hence, lakes play an important role within the terrestrial carbon budget.

Wetlands and lakes are particularly abundant in the subarctic and boreal regions (Smith et al., 2007; Verpoorter et al., 2014), where climate warming is occurring at a faster pace than in the rest of the world (Serreze and Barry, 2011). In this context, freshwaters have received increasing attention over the past decade, due to the potential for lakes and particularly

Arctic thermokarst lakes to exert a feedback on climate warming through large $CH_4$ emissions (Walter Anthony et al., 2016; Walter et al., 2006). While non-thermokarst, post-glacial lakes emit less $CH_4$ per unit area (Sepulveda-Jauregui et al., 2015; Wik et al., 2016) they cover a larger area and may as a whole emit half of the total $CH_4$ emissions (16.5 +- 9.2 Tg $CH_4$ $yr^{-1}$) recently attributed to northern (> 50 N) lakes and ponds (Wik et al., 2016).

Biogenic $CH_4$ production (methanogenesis) occurs in anoxic environments such as lake sediments and water-saturated peat

(Cicerone and Oremland, 1988). The process is controlled by the interplay between temperature and the input of organic matter (Kelly and Chynoweth, 1981; Yvon-Durocher et al., 2014; Zeikus and Winfrey, 1976). In lakes and waterlogged wetlands, $CH_4$ reaches the atmosphere from its production zone via direct bubble release up to the surface (ebullition), through emergent vascular plants, or via turbulence-driven diffusion through the water column (Bastviken et al., 2004; Lai, 2009; Rudd and Hamilton, 1978). The net flux of $CH_4$ at the lake surface is a balance between the production in the

sediments and the oxidation of $CH_4$ into $CO_2$ at oxic/anoxic boundaries within the water column (Casper, 1992). In shallow lakes, ebullition is a main pathway for $CH_4$ to reach the atmosphere while bypassing the oxidation zones (Bastviken et al., 2004). In wetlands, transport from the peat soil to the surface through vascular plants by passive diffusion or by pressurization effects depending on the plant species (Brix et al., 1992), is an efficient pathway for $CH_4$ to avoid oxidation in the soil and water before reaching the atmosphere (Joabsson and Christensen, 2001).

Dissolved $CO_2$ in lake is produced throughout the water column and sediments (Casper et al., 2000) or is directly imported from the catchment (Maberly et al., 2013; Weyhenmeyer et al., 2015). *In situ* production comes from the mineralization or the photochemical oxidation of carbon (C) input from the surrounding catchment (Cory et al., 2014; Dillon and Molot, 1997;





Duarte and Prairie, 2005) and from the degradation of locally produced organic carbon. $CO_2$ exchange across the air-water interface is primarily via diffusive release rather than ebullition (e.g. Casper et al., 2000). Lake waters are generally observed to be supersaturated in $CO_2$ with respect to atmospheric values due to in-lake respiration processes outweighing rates of primary production (Duarte and Prairie, 2005; Sobek et al., 2005). Hence they are generally $CO_2$ sources to the

atmosphere, albeit nutrient-rich lakes and ponds can be small $CO_2$ sinks during summer months (Huotari et al., 2011) or an entire summer season (Laurion et al., 2010; Pacheco et al., 2013; Shao et al., 2015; Striegl and Michmerhuizen, 1998; Tank et al., 2009).

Near-surface atmospheric forcing is a key driver for the transport and net emissions of gases from a lake to the atmosphere. Ebullition of $CH_4$ in lakes is partly triggered by water level changes or drops in atmospheric pressure (Casper et al., 2000;

Mattson and Likens, 1990), as a decrease in the hydrostatic pressure of the overlying water column on gas saturated sediments favors the release of bubbles (Varadharajan and Hemond, 2012). Wind-driven turbulence is a recognized driver of diffusion-limited exchange of $CO_2$ and $CH_4$ across the lake-water interface (Sebacher et al., 1983; Wanninkhof et al., 1985). Convective mixing due to the cooling of the lake surface or following the breakdown of thermal stratification in the water column can increase advection of gas-rich water from the lake bottom thus enhancing the diffusion-limited release of

gases to the atmosphere (Eugster, 2003; MacIntyre et al., 2010; Podgrajsek et al., 2015), especially if the main source of those gases is the sediments as is the case for $CH_4$.

Northern ecosystems have strong seasonal contrasts with short growing seasons and long snow-covered winters. A snow cover insulates the soil from very cold air temperature (Bubier et al., 2002), while an ice lid on a lake temporarily inhibits the exchanges of gas and heat between the water and the atmosphere (e.g. Greenbank, 1945) and greatly dampens wind-

driven turbulence in the water column. In lakes, gases can accumulate at the bottom during long stratification periods, or under lake ice. Lake overturn events most often occur in spring for seasonally ice-covered lake and/or at fall if lakes thermally stratify during summer (Kirillin et al., 2012; Wetzel, 2001). Overturn events can thus lead to the fast release of accumulated gas to the atmosphere (Jammet et al., 2015; Michmerhuizen et al., 1996; Phelps et al., 1998) but also to the input of atmospheric oxygen, thus increasing the potential for methanotrophy (Kankaala et al., 2006; Schubert et al.,

2012). Extension of observations across all seasons of the year is rare in high northern latitudes, particularly for lakes, yet indispensable to reduce the uncertainty in the magnitude of annual carbon exchange and to improve understanding of the processes driving them.

Common methods to measure lake-atmosphere fluxes include floating chambers (e.g. Bastviken et al., 2004), gas transfer models (e.g. Cole and Caraco 1998), and bubble traps (e.g. Walter et al., 2008; Wik et al., 2013; Sepulveda-Jauregui et al.,

2015). If these methods are not integrated and combined, they inherently omit part of the total surface flux. The application of the eddy covariance (EC) method (Aubinet et al., 2012; Moncrieff et al., 1997) to lake environments (e.g. Anderson et al., 1999; Eugster, 2003; Huotari et al., 2011; Mammarella et al., 2015; Podgrajsek et al., 2014; Shao et al., 2015) offers



long-term flux monitoring and is a potential methodological solution for solving the spatial and temporal issues of measuring total gas exchange in lakes. A few studies, so far, have used eddy covariance to quantify long-term $CO_2$ emissions from boreal lake (Huttunen et al 2011) as well as $CH_4$ emissions from boreal (Podgrajsek et al., 2014) or subarctic lakes (Jammet et al., 2015). We report here one of the first year-round eddy covariance measurements of both $CH_4$ and $CO_2$ fluxes from a

seasonally ice-covered lake.

Surface fluxes were monitored using the eddy covariance method in a subarctic permafrost peatland undergoing thaw, a landscape with a high percentage of pond and lake coverage. The location of the flux tower allowed for measurements to alternate between surface fluxes from a shallow lake and from a permafrost-free, waterlogged fen-type wetland. The overall aim of this work was to quantify year-round carbon fluxes in a post-glacial lake, a type widely present around the subarctic,

as compared to the adjacent fen. Specifically, the objectives were (1) to compare the seasonality of $CH_4$ and $CO_2$ fluxes from two contrasting subarctic ecosystems (lake and fen), (2) to explore the possibility of identifying short-term environmental controls on the surface-atmosphere exchange of $CH_4$ and $CO_2$ in a lake, using high, sub-daily temporal resolution measurements covering all seasons of the year and (3) to assess and compare the annual atmospheric carbon budget of a lake and a wetland.

## 2. Materials and methods

### 2.1 Study site

Stordalen Mire is a subarctic peatland complex (68°20' N, 19°03' E) with a high lake and pond coverage, located near Abisko in Northern Sweden. Mean annual temperature in the Abisko region has been increasing and fluctuating around 0°C

since the 1990's as part of an accelerated warming trend (Callaghan et al., 2010). Permafrost, which is discontinuously present in the local mires, has been thawing at an increased rate since the 1990's in the peatlands of the region, sometimes disappearing completely (Åkerman and Johansson, 2008). In Stordalen Mire permafrost thawing has led to changes in microtopography, which controls local hydrology, which in turn leads to vegetation shifts (Christensen, 2004; Malmer et al., 2005). This accentuates the heterogeneity of a landscape comprised of elevated palsas with permafrost, thawing lawns

with thermokarst ponds, and permafrost-free, water-saturated fens. The mire is bordered by post-glacial lakes on its western, northern and eastern edges. This study focuses on the lake Villasjön (Fig. 1) and the adjacent fen to the west of the tower. The two ecosystems may be hydrologically connected with a directional flow from the lake to the fen (Olefeldt and Roulet, 2012).

The wetter fen areas have expanded as a result of permafrost thaw over the past decades (Johansson et al., 2006). The

water table of the fen is at or above the surface throughout the year. The dominant vegetation species are vascular plants



*Carex rostrata* and *Eriophorium angustifolium*. Villasjön is the largest (0.17 km$^2$) lake of the 15 km$^2$ wide Stordalen catchment containing 27 lakes (Lundin et al., 2013). It has a mean and maximum depth of 0.7 m and 1.3 m, respectively (Jackowicz-Korczyński et al., 2010; Wik et al., 2013). The upstream catchment of Villasjön is dominated by birch forest (Olefeldt and Roulet, 2012), while its western shore is bordered by thawing palsa. There is a vernal, small surface inflow

feeding Villasjön on the east during snow melt (Wik et al., 2013) but the lake can be considered a hydrologically closed system in winter (Boereboom et al., 2012) and usually freezes to the bottom. The DOC concentration in the lake water has been measured to be 8.1 mg l$^{-1}$ in 2008 (Olefeldt and Roulet, 2012). Aquatic vegetation is present in the lake, on its bottom as algae, as submerged plants within its southern arm and in low density of emergent macrophytes at its shores.

## 2.2 Eddy covariance measurements

### 2.2.1 Measurement set-up

Between June 2012 and December 2014, the surface–atmosphere exchange of $CH_4$, $CO_2$ and sensible heat flux has been monitored nearly continuously with an eddy covariance set-up located at the shore of Villasjön (Fig. 1). The 2.92 m high mast was equipped with a 3-D sonic anemometer (R3-50, Gill Instruments Ltd.) sampling wind components and sonic temperature at 10 Hz. Throughout the study period, ambient molar densities of $CO_2$ and $H_2O$ were sampled at 10 Hz with an

open path infrared gas analyzer (IRGA), model LI7500 (LICOR Environment, NE, USA) mounted on the mast at 2.50 m height. Following lightning that hit the electric grid in Stordalen Mire on 27 July 2013, the initial LI7500 ceased functioning and was replaced on 1 October 2013 by a different instrument of the same model.

From June 2012 to May 2013, ambient $CH_4$ mixing ratio was sampled at 10 Hz with a closed path Fast Greenhouse Gas Analyzer (FGGA, Los Gatos Research, CA, USA) in air that was taken from a gas inlet located at 2.50 m on the mast

through a 6 mm inner diameter polyethylene (PE) tube using a dry scroll pump (Varian TriScroll 300). The efficient flow rate in the 95 m long sampling line was 16 L min-1, ensuring the maintenance of turbulent conditions (Reynolds number ca. 4025). On 5 June 2013, the tube was replaced with an 8 mm inner diameter PE tube, which changed the flow rate in the sampling line to 23.88 L min-1 (Reynolds number ca. 4099). On 4 August 2013, the closed path $CH_4$ system was renewed: the IRGA was changed to a FGGA model 911-010 (Los Gatos Research, CA, USA). Due to instrumental maintenance, the

IRGA was offline between February and March 2014 and replaced on 24 March 2014 by the previous FGGA. From August 2013 to December 2014, due to a failure in the electronic connection between the gas analyzer and the data logger, the raw $CH_4$ data were pre-processed in order to align the timestamp and frequency of the gas analyzer recordings with the time stamp of the logger sampling the wind data. Each change in the $CH_4$ measurement set-up was taken in account in the flux calculation.



### 2.2.2 Flux calculation and quality-check

$CO_2$, $CH_4$, sensible and latent heat fluxes as well as atmospheric turbulence quantities were calculated and output as 30 min averages using the open source software EddyPro version 5.2. (hosted by LICOR Environment, USA). Processing of the 10 Hz raw data followed standard eddy covariance procedures (Aubinet et al., 2012; Lee et al., 2005); methodological choices differed partly for $CH_4$ and $CO_2$ flux processing as detailed in Appendix A.

The 30min averaged fluxes were quality checked to detect measurement errors and to ensure the fulfillment of theoretical assumptions for the application of the eddy covariance method. Spikes were present in the $CO_2$ flux time series, which can be due to weather conditions, to fast changes in the atmosphere's turbulent conditions or to faulty instrumentation. Outliers in the $CO_2$ flux dataset were detected using the median of absolute deviation from the median (MAD) as described in Papale et al. (2006) using a threshold z = 4 and with no distinction between day and night. Additionally, $CO_2$ flux averaging periods were rejected when the number of spikes per half hour was > 100 (Mammarella et al., 2015) and when skewness and kurtosis were outside the [-2,2] and [1,8] ranges (Vickers and Mahrt, 1997), respectively. Required flux stationarity (FST) within the 30 min flux averaging period was ensured by rejecting each flux value when the FST criterion as defined by Foken and Wichura (1996) was above 0.3. The fen part of the $CO_2$ flux dataset was additionally filtered for poorly developed turbulence (Mauder and Foken, 2006) and $u*$-filtered with a threshold determined to be 0.1 m s-1 (Papale et al., 2006).

Quality-check and screening of the $CH_4$ flux time series included rejection of averaging periods when the number of spikes per half hour was > 100. Fluxes were rejected when skewness and kurtosis were outside the pre-cited range, in winter only, due to the pulse-character of $CH_4$ emissions during non-winter periods that could be misinterpreted as faulty raw data due to a high skewness (Jammet et al., 2015). Flux stationarity within the averaging time period was also ensured by rejecting values when the FST criterion was above 0.3. Additionally, flux values for which a time lag was not found within the plausibility time lag window were rejected (Eugster et al., 2011; Wille et al., 2008). Gaps were present initially present in the flux dataset due to instrument malfunctioning and power cuts. After quality-check and filtering, in total 13474 $CH_4$ fluxes and 11629 $CO_2$ fluxes were available for further analysis (Table S1). Data rejection rate was high; eddy covariance studies in lake environments usually report high rejection rates from quality-check routines (Jonsson et al., 2008; Mammarella et al., 2015; Nordbo et al., 2011)

### 2.3 Flux footprint and partitioning between lake and wetland

As in Jammet et al. (2015), the bidirectional wind pattern at the Stordalen Mire was used to partition the flux dataset into two main wind sectors which crossed the two different studied ecosystems. To ensure a clear distinction between the lake and the fen fluxes and the homogeneity of each surface, the lake sector was conservatively defined as 20° - 135°, and the fen



sector as 210° - 330° (0° = 360° is true north). Over the study period, 46.2 % of the measured fluxes originated from the lake sector, 45.7 % of the measured fluxes originated from the fen sector, and the residual 8.1 % of the measured flux data was excluded, for being of lower wind speed and originating from mixed sources. Each season was nevertheless well represented in the flux dataset of each ecosystem (fen and lake) thanks to regular shifts in wind direction.

The flux footprint was calculated using a 2D model developed by Kljun et al. (2015). Inputs to the model include turbulence quantities measured at the tower (friction velocity ($u^*$), standard deviation of cross-wind velocity ($\sigma_v$), Obukhov length, horizontal wind speed), height of the boundary layer (derived from the Era-Interim reanalysis product, (Dee et al., 2011)), the measurement height and surface roughness length which was separately estimated for the lake and the fen sectors. Footprints were calculated for each 30 min time step where the required data was available, and averaged for the periods of
interest (summer and winter).

According to the footprint the model, most of the flux measured at the tower (peak fetch) originated from a distance of 38 m on the fen sector and 73 m on the lake sector, on average during the ice-free season. The lake surface was within the cumulative 80 % of flux footprint during the ice-free seasons (Fig. 1). In winter, the footprint model revealed that the 80 % cumulative footprint, on the lake side, included part of the land in the middle of the lake (Fig. S1). Thus, to avoid large
contamination of the land respiration in the winter lake fluxes due to an extended footprint over the lake shores, the winter fluxes from the lake sector were only kept for further analysis when the standard deviation of lateral wind velocity $\sigma_v \leq 1$ m s$^{-1}$. This removed an additional 3.7 % of the lake $CO_2$ flux data set.

### 2.4 Ancillary measurements

Supporting environmental variables were measured at 1 Hz and averaged and logged every half hour. Temperature probes
(T107, Campbell Scientific, Inc., UT, USA) were installed in the peat 5 m south-west from the tower at 5 cm, 10 cm, 25 cm, and 50 cm depths. Net radiation at the fen surface was recorded with an REBS Net Radiometer, model Q7.1 (Campbell Scientific, Inc., UT, USA). Air temperature used in this study was measured at 2 m height on a mast located in the middle of the lake (Fig. 1) with a CS215 probe (Campbell Scientific, Inc., UT, USA). At the same mast, radiation components were measured with a CNR4 Net Radiometer (Kipp & Zonen, The Netherlands) from which net radiation at the lake surface was
computed. Water temperature in the center of the lake (Fig. 1) was measured at 10 cm, 30 cm, 50 cm, and 100 cm depths with intercalibrated HOBO Water Temp Pro v2 loggers (Onset Computer Corporation, MA, USA). The loggers are suspended on a nylon line from a mooring float, which stays at the surface throughout the year. The string assembly was designed so that the bottom sensor at 100 cm depth is in the surface sediment. Lake surface albedo was computed daily as the midday ratio (10:00 to 13:00) of the shortwave radiation components, and used in combination with temperature data to
delimit the ice cover seasons. Air pressure and precipitation were measured at a weather station 640 m south from the eddy covariance mast.





## 2.5 Gap filling of $CH_4$ and $CO_2$ flux time series

The estimation of annual carbon exchange budgets required filling the gaps in the $CH_4$ and $CO_2$ flux time series. When working with highly skewed flux datasets (Fig. C1), integrating the mean flux over the whole time period may lead to important overestimation. High frequency $CO_2$ and $CH_4$ flux measurements in lake ecosystems are still rare. There is no
published account of gap filling $CH_4$ fluxes measured with eddy covariance, and relationships between high resolution fluxes and environmental variables are not yet well characterized.

Gap filling of the $CH_4$ flux time series was performed separately on the lake and fen flux datasets, using artificial neural networks (ANN) (Moffat et al., 2010; Papale and Valentini, 2003). Artificial neural networks (ANNs) are multivariate, non-linear regression models that are fully empirical: the observational data are used to constrain the model's numerical
relationship between the inputs (independent variables i.e. environmental drivers) and output (dependent variable i.e. fluxes) (Moffat et al., 2010). ANNs have been tested and successfully used to estimate missing values and gap fill $CO_2$ flux time series measured with eddy covariance in forests (Moffat et al., 2007; Papale and Valentini, 2003) or in urban terrain (Järvi et al., 2012) and to gap fill $CH_4$ emissions in wetlands (Dengel et al., 2013).

Three ANN models were built separately on $CH_4$ emissions from the fen, $CH_4$ emissions from the lake and $CO_2$ emissions
from the fen, on the hourly scale. Model development followed workflows introduced by Papale and Valentini (2003), Moffat et al. (2007) and Dengel et al. (2013). Environmental variables to be included as inputs were selected according to their physiological relevance to the production of $CH_4$ and $CO_2$ and their transport from the surface to the atmosphere, as reported in the literature. A detailed description of model development and list of input variables can be found in supplementary Text S1 and Table S2. Gaps in the measured $CH_4$ and $CO_2$ flux time series were then replaced by predicted
values, and the annual sums were computed by integrating the hourly flux values over time. The performance of the ANN models was assessed by comparing the predicted values with original observed values over the whole dataset. The goodness of fit was quantified with the coefficient of determination $r^2$ and the absolute root mean square error (RMSE). The ANN gap filling was most performant on the fen $CH_4$ flux dataset (Appendix B). The predictive performances of the models depended on the amount of gaps in the original dataset and on the accuracy of the choice of environmental drivers.

The ANN method did not perform as well on the lake $CO_2$ flux time series. Thus the seasonal lake $CO_2$ exchange was computed by multiplying the mean flux rate during each season per the number of days. Considering the normal distribution of the $CO_2$ lake fluxes (Fig. C1), this method was considered an acceptable way of filling missing values. In the case of a highly positively skewed dataset such as the $CH_4$ fluxes this would lead to a large positive bias and overestimation of the annual sum.



## 2.6    Uncertainty analysis

The total random error is a composite of errors associated with instrument noise, the stochastic nature of turbulence, the instrument precision and the variation of the flux footprint (Moncrieff et al., 1996). The absolute random error increases with the magnitude of the flux (Richardson et al., 2006), while over time it decreases with increasing size of the dataset

because of its random nature (Moncrieff et al., 1996). It is therefore negligible when propagated over annual sums but can be important for single flux values. The random error was calculated as the sampling error for each flux value in the flux calculation software using the method of Finkelstein and Sims (2001). Considering that the total flux error of EC gives an estimation of the value under which a flux cannot significantly be differentiated from noise (the precision of the measurements), we considered this error estimation as a detection limit of the flux measurement system.

The mean random error of the quality-checked measured $CH_4$ fluxes was $2.9 \pm 4.3$ (mean ± STD) nmol m$^{-2}$ s$^{-1}$ at the lake and $4.7 \pm 3.8$ nmol m$^{-2}$ s$^{-1}$ at the fen, which corresponds to 7.6 % and 6 % of the overall mean measured fluxes, respectively. This is in the lower end of ranges reported for $CO_2$ and energy fluxes. The mean random error of the individual measured lake $CH_4$ fluxes was highest in winter (24 %) and lowest during ice-out (11 %) when the largest fluxes were measured. The mean random error of the quality-checked measured $CO_2$ fluxes was $0.20 \pm 0.57$ µmol m$^{-2}$ s$^{-1}$ at the lake and $0.32 \pm 0.48$ µmol m$^{-2}$

s$^{-1}$ at the fen, equivalent to 31 % and 13 %, respectively, of the overall mean absolute measured flux, indicating that our set-up was able to measure both $CH_4$ and $CO_2$ fluxes observable at our site. These values are comparable to what has been reported in vegetated (Finkelstein and Sims, 2001), urban (Järvi et al., 2012), boreal lake (Mammarella et al., 2015) and other typical eddy covariance sites (Rannik et al., 2016). In spring the random error of $CO_2$ fluxes at the lake was 18 % of the measured flux, however during the ice-free season it was above the absolute measured flux in 20 % of the cases. Thus

air-lake $CO_2$ exchange rates during the summer were low and sometimes close to the detection limit.

The random uncertainty of the fluxes modeled with ANN was on average 4 nmol m$^{-2}$ s$^{-1}$ for fen $CH_4$ fluxes; 0.23 µmol m$^{-2}$ s$^{-1}$ for fen CO2 fluxes; 11 nmol m$^{-2}$ s$^{-1}$ for lake $CH_4$ fluxes. These errors were small when propagated onto seasonal and annual sums using random error propagation principle (Moncrieff et al., 1996). The systematic bias on the annual flux due to the gap filling method, the bias error (BE), was calculated on the seasonal and annual flux sums as in Moffat et al (2007) as the

sum of the difference between the predicted values $p_i$ and the observed values $o_i$:

$$BE = \frac{1}{N}\sum(p_i - o_i) \qquad\qquad (1)$$

where N is the number of gap filled values in the flux time series. The bias error adds up over time. It was multiplied by the number of gap filled values to obtain a total seasonal and annual offset (Moffat et al., 2007). The offset can be the largest source of uncertainty in the computation of annual budgets. It is dependent on the accuracy of the model but also on the

amount of gaps in the original dataset to fill. The systematic offset due to gap filling was larger for annual $CO_2$ fen fluxes (Table 3), likely because of the higher noise in the measurements and higher amount of gaps during the second year. For



total lake $CO_2$ fluxes, the annual BE was calculated in the same way, using seasonal mean as the predicted values. The resulting offset due to gap filling was close to zero (Table 3), because flux values were normally distributed, which confirms that using the mean was not inducing a significant bias for this particular dataset.

## 2.7 Definition of seasons

Observations started in June 2012 and were sustained nearly continuously until December 2014. We adopted a lake-centric definition of seasons based on the lake ice phenology. A full year was defined from 1 June to 31 May of the next year, so that an entire ice-cover season was included in a given year. This keeps the connected thaw period and previous ice-cover season within the same year. The year was further divided into an ice-cover season (winter), an ice-free season (summer and fall) and a thaw season (spring). The thaw season represents a transitional period during which the snow cover and lake ice

melt. It is separated from summer, since the hydrological and biogeochemical dynamics in seasonally ice-covered lakes differ from the rest of the open water season.

Daily air temperature was used to define seasons. The start of the ice-cover season was defined as the start of the lake freeze up, i.e. the first day on which daily mean air temperature is below zero for three consecutive days. Further, this date coincided each year (2012 to 2014) with the formation of thermal stratification in the water column (Fig. 2d) due to a rise in

bottom water temperature right after the first day of freezing (Fig. 2a, blue line). This indicates the inhibition of direct heat exchange between the lake and the atmosphere when ice forms at the lake surface. In the first year (2012), the first day of the ice-cover season was confirmed by visual observation of ice over the whole lake surface while the following years, these combined temperature observations were used to define the start of ice cover.

The end of the ice-cover season was defined as the start of thaw, i.e. the first date on which daily mean air temperature rose

above 0°C for at least three consecutive days. This date preceded by 1 to 2 days the temperature rise to 0°C in the surface (10 cm) of the lake ice. Further, the start of the thaw season could be confirmed by the increase of daily energy input (upwelling > downwelling radiation) to net positive values at the lake and the fen surfaces (Fig. 2b) and by a decrease in lake albedo (Fig. 2c) until reaching open water values (mean = 5 %). We thereby defined two complete ice-cover seasons during the study period: from 15 October 2012 to 14 April 2013 (182 days) and from 16 October 2013 to 10 April 2014 (177 days). In

2014, freeze up of the lake occurred on 10 October. The ice-cover seasons were systematically characterized by negative daily energy input at the lake and at the fen (Fig. 2). Thus our season definition based on daily air temperature was robust and reproducible each year.

The end of spring and beginning of the ice-free season was defined as the first date with a daily temperature gradient in the water column close to 0°C m$^{-1}$ after the spring overturn, i.e. when the lake enters its isothermal conditions after complete ice





thaw. The dataset of this study covers two complete years plus an additional ice-free season, i.e. three ice-free seasons, two ice-cover seasons and two springs (ice-out seasons). Seasons dates and lengths are summarized in Table 1.

## 3. Results

### 3.1 Environmental conditions and lake climatology

Mean annual (June to May) air temperature measured at the Abisko Scientific Research Station was -0.3°C in 2012-2013 and 0.9°C in 2013-2014. The latter was significantly above the long-term average (1913-2014) of -0.5 ± 1°C (mean ± STD). The ice-free season was warmest in 2014 and coolest in 2012 (Table 1). This difference between years was reflected both in the mean daily peat temperature measured at 10 cm depth and in the mean daily lake water temperatures, which were warmest in 2014 (Figure 2; Table 1). The differences in mean annual temperature observed over our study period coincide with an increase in total net radiation at the lake and fen surfaces both annually and during the ice-free period during 2012 to 2014 (Table 1). The winter 2013-2014 was on average 2.1°C warmer than the previous one and the ice-cover season was 5 days shorter due to an earlier thaw start. The thaw period started earlier in 2014 but lasted longer (Fig. 2d, Table 1); complete lake overturn (isothermal water column) following ice thaw occurred 7 days later than in 2013.

The development pattern of thermal stratification along lake depth at freeze up and its breakdown in spring was repeated each year (Fig. 2d). Both in spring 2013 and 2014, the lake overturn occurred after the development of large thermal stratification along lake depth during ice-out (Fig.2d). Temperature at the bottom of the lake was up to 4°C warmer than the surface in spring 2013 and up to 6°C warmer in 2014 likely indicating the penetration of solar radiation through thinning ice before complete ice out and full water mixing. The thermal structure at the beginning of freeze up, as well as during the period of ice thaw preceding spring overturn in 2013, has been previously described in detail for this site in a winter-focused study (Jammet et al., 2015).

During the ice-free, summer season, there was slight to no thermal stratification in the water column (Fig. 2d). The shallow lake water column reacted quickly to temperature changes (Fig. 2a,d) and the dark bottom warmed up fast since solar radiation can reach the sediment surface. Thus, the lake had a polymictic behavior, i.e. it was regularly mixing to the bottom during the ice-free season, which ensures isothermal conditions in the water column throughout the summer. Water temperature reached a maximum of 23.8°C in July 2014. Conversely, a strong temperature gradient formed during winter (Fig. 2f). In both winters the surface sediment temperature dropped below 0°C by February, with minima of -1.9°C in March 2013 and -1.6°C in February 2014. This suggests that the water column froze to the bottom. The temperature in the peat soil and at the surface sediment in the lake were de-coupled from air temperature (Fig. 2a) showing the hindrance of heat exchange with the atmosphere due to the presence of ice and snow at the surface..



## 3.2 Year-round $CH_4$ and $CO_2$ fluxes

$CH_4$ emissions from the lake followed a highly skewed distribution (Fig. C1a); there was a large difference between the overall mean (40 nmol $m^{-2}$ $s^{-1}$) and overall median (12 nmol $m^{-2}$ $s^{-1}$). The seasonal flux pattern was characterized by low background emissions and occasional, large degassing events (Fig. 3) with 25 % of measured data above 111 nmol $m^{-2}$ $s^{-1}$ in

the ice-out period, and 5 % of measured data above 75 nmol $m^{-2}$ $s^{-1}$ within the ice-free season (Fig. 4). Over the full measurement period, summer emission rates averaged to 26 nmol $m-2$ $s-1$ (median 12 nmol $m-2$ $s^{-1}$), ice-out emission rates to 84 nmol $m^{-2}$ $s^{-1}$ (median 33 nmol $m^{-2}$ $s^{-1}$) and winter emissions were at 2.8 nmol $m^{-2}$ $s^{-1}$ (median 3.0 nmol $m^{-2}$ $s^{-1}$). The mean and median of the winter emissions were not significantly different from the mean random error of the fluxes. $CO_2$ fluxes were close to normally distributed at the lake (Fig. C1b) and the overall mean rate was 0.22 µmol $CO_2$ $m^{-2}$ $s^{-1}$ (median

0.18 µmol $m^{-2}$ $s^{-1}$). There was a distinctive $CO_2$ outgassing at the time of water overturn during the spring of both 2013 and 2014 (Fig. 3d). The mean measured $CO_2$ exchange at the lake was significantly negative during the ice-free seasons (one sample t-test, p < 0.001), -0.14 µmol $m-2$ $s-1$, indicating a low but net uptake of $CO_2$ (Fig. 4). Negative flux rates started right after the spring $CO_2$ outgassing in 2013 and 2014 (Fig. 3d). The highest $CO_2$ uptake rates were observed during the summer of 2014, which was the warmest summer of the study period with highest solar radiation input (Table 1). In fall

2014, a burst of $CO_2$ was measured at the lake, which was not present in previous years (Fig. 3d). This fall burst was not observed in the $CH_4$ flux measurements. Wintertime $CO_2$ emissions from the lake were significantly above the flux random error and significantly positive (Fig. 3d, Fig. 4). There was an inter-annual variability in the magnitude of the $CH_4$ spring degassing from the lake between the two years (Fig. 3c). In contrast, mean $CO_2$ degassing was higher during the second ice-out period, 0.78 µmol $CO_2$ $m^{-2}$ $s^{-1}$ (median 0.47 µmol $CO_2$ $m^{-2}$ $s^{-1}$) in 2013 and 0.99 µmol $CO_2$ $m^{-2}$ $s^{-1}$ (median 0.75 µmol

$CO_2$ $m^{-2}$ $s^{-1}$) in 2014. Both $CH_4$ and $CO_2$ emissions during ice-out were significantly higher than during the following ice-free season, in both years (Fig. S2).

The distribution of $CH_4$ emissions from the fen was less skewed (Fig. C1a) and the overall measured mean was 77 nmol $m^{-2}$ $s^{-1}$ (median 58 nmol $m^{-2}$ $s^{-1}$). The highest $CH_4$ emissions at the fen occurred during the ice-free season (Fig. 3), with a mean rate of 110 nmol $m^{-2}$ $s^{-1}$ (median 108 nmol $m^{-2}$ $s^{-1}$). Sustained $CH_4$ emissions were measured at the fen throughout the winter

(Fig. 3), with a mean rate of 25 nmol $m^{-2}$ $s^{-1}$ (median 25 nmol $m^{-2}$ $s^{-1}$). Flux rates during the snow-melt and ice-out season averaged to 35 nmol $m^{-2}$ $s^{-1}$ (median 33 nmol $m^{-2}$ $s^{-1}$). $CO_2$ fluxes from the fen averaged to -1.3 µmol $m^{-2}$ $s^{-1}$ (median 0.2 µmol $m^{-2}$ $s^{-1}$) over the full period and the mean rate was -2.6 µmol $m^{-2}$ $s^{-1}$ during the ice-free seasons. $CO_2$ respiration was sustained throughout winter (Fig. 3b) at a mean rate of 0.8 µmol $m^{-2}$ $s^{-1}$ and the release of $CO_2$ during the melt season was low (Fig. 4).

Average annual seasonality of $CH_4$ and $CO_2$ fluxes in both ecosystems is shown in Fig. 5. The lake dominated $CH_4$ and $CO_2$ emissions during spring. During the ice-free seasons, on the contrary, the lake was a lower emitter of $CH_4$ than the fen per unit area, and its $CO_2$ exchange was close to neutral with a small uptake. There was a slight seasonality in $CH_4$ lake




emissions during summer but, annually, $CH_4$ and $CO_2$ lake fluxes peaked in spring (Fig. 5). $CH_4$ emissions from the fen peaked in August and net $CO_2$ exchange peaked in July (Fig. 5).The emission of both gases occurred at lower rates but continuously in winter.

### 3.3 Variability of lake-air carbon exchange within seasons

The ice-out and ice-free periods are different in terms of flux dynamics. The dataset was therefore separated into seasons to explore controls on the lake fluxes. During the ice-free seasons, half-hourly $CH_4$ fluxes at the lake were characterized by degassing events that coincided with drops in atmospheric pressure (Fig. 6). Daily EC flux data were compared with spatially averaged, daily ebullition fluxes measured in the lake with ebullition traps located nearby or within the tower footprint. The degassing events measured with the eddy covariance system coincided in timing and in magnitude with the

daily ebullition fluxes measured with the ebullition traps (Fig. 6).

Due to the high skewness of the $CH_4$ lake flux dataset (Fig. C1) and the presence of outliers in both $CO_2$ and $CH_4$ flux datasets from the lake, we used Spearman's $\rho$ coefficient, which is a statistical measure of association that is robust to outliers and applicable to skewed distribution (Kowalski, 1972), to explore bivariate associations. Among potential flux drivers, the highest correlation of ice-free $CH_4$ emissions was found with surface sediment temperature ($\rho = 0.49$, Table 2).

Degassing events occurred most of the time following an increase in surface sediment temperature (Fig. 6). When averaging all 30min $CH_4$ fluxes during the three ice-free seasons per bins of 1˚C, an exponential regression between lake $CH_4$ fluxes and surface sediment temperature could explain 82 % of the variability in $CH_4$ emissions (Fig. 7).

Wind speed correlated best with $CH_4$ lake emissions during the ice-out period (Table 2, $\rho = 0.40$). The relationship was weaker during the open water period. We observed a few $CH_4$ degassing events in summer that coincided with the likely

mixing of the water column following a short period of thermal stratification (0.8 to 1˚C gradient between 10 cm and 100 cm depths, Fig. S3a,b) but these were not systematic (Fig. S3c,d). During the ice-free season, there was a weak anti-correlation between $CO_2$ exchange at the lake and air and water temperature (Table 2). Wind speed correlated weakly with $CO_2$ lake exchange, while there was a strong anti-correlation with sensible heat flux at the lake surface (Table 2), and with solar radiation input (Table 2).

During the ice-out seasons, half-hourly $CO_2$ fluxes strongly correlated with $CH_4$ emissions ($\rho = 0.67$, p < 0.001, Table 2) and followed the same emission pattern at the half-hourly scale, both in 2013 and 2014 (Fig. S4). This correlation was not sustained during the open water seasons, when the two flux datasets had a very different short-term variability. Both $CH_4$ and $CO_2$ degassing in spring positively correlated with increasing air temperature.

The diurnal course of $CO_2$, $CH_4$ and turbulent energy fluxes at the lake and the fen was calculated on hourly fluxes. Only

days with more than 75 % of hourly data coverage were selected. The median flux of each hour was then computed across



all days within each year during the open-water season, and plotted along with the 25[th] and 75[th] percentiles (Fig. 8). There was no diel cycle visible in the fen $CH_4$ fluxes, while net $CO_2$ exchange at the fen surface showed a clear peak uptake at noon. Lake methane fluxes tended to be higher fluxes in the morning hours. A systematic diel pattern was observed in the lake $CO_2$ fluxes during each open water season (Fig. 8), with a slight peak in the mornings in 2012 and 2013 while $CO_2$ uptake peaked in the middle of the day in 2014. The 24-hour cycle of sensible heat flux (H) at the lake peaked in the late morning (Fig. 8) at ca. 10 a.m. local time (median 20 W m$^{-2}$), while being < 10 W m$^{-2}$ in the afternoon. The diel $CO_2$ pattern at the lake also coincided with daily variation in water surface and air temperature and was in antiphase with the diel pattern of sensible heat flux. Latent heat flux (LE) at the lake peaked in the afternoon. At the hourly time scale, solar radiation could explain 88 % of the diel variability in air-lake $CO_2$ exchange during the summer months (Fig. 9). This light response curve resembled the one measured at the fen, although less pronounced.

### 3.4 Annual atmospheric carbon budget

We report seasonal emissions for all available seasons, and for lake $CO_2$ total annual emissions for the first year only (June 2012 to May 2013), due to the absence of a robust gap filling model for the second year, which had least data coverage. During the ice-free seasons, total lake $CO_2$ flux was negative and total lake $CH_4$ flux was positive (Table 3). Annually, the largest contribution to total carbon exchange at the lake was from the ice-out season, whereas the ice-free season was quantitatively the most important period for the annual carbon exchange at the fen (Table 3). On average over both years, the ice-cover season accounted for 33 % of the fen annual carbon ($CH_4$+$CO_2$) exchange per m$^2$. During the first year, the lake C-emissions equaled 70 % of the total net fen C-exchange. On a carbon mass basis, $CO_2$ exchange dominated the total carbon budget. Total net annual carbon exchange ($CH_4$+$CO_2$) at the fen was -38.2 g C m$^{-2}$ in the first year and -52 g C m$^{-2}$ in the second year (average -45.1 g C m$^{-2}$ yr$^{-1}$), while the lake total carbon exchange was positive at 26.7 g C m$^{-2}$ yr$^{-1}$ (first year only), of which 80 % was emitted as $CO_2$.

## 4. Discussion

### 4.1 Contrasting annual seasonality of carbon fluxes between lake and fen

The average annual seasonality of the fluxes across the study period (Fig. 5) shows that both ecosystems had different peak timings in terms of $CH_4$ and $CO_2$ exchange. $CH_4$ emissions from the fen followed the expected seasonality of emissions from boreal and subarctic wetlands (Hargreaves et al., 2001; Jackowicz-Korczyński et al., 2010; Rinne et al., 2007). The dense emergent vegetation cover at the waterlogged fen dominated by vascular plants, which are efficient conduits for $CH_4$ to reach the atmosphere,  lead to maximum primary production of organic carbon during the summer. The continuous emission of




$CH_4$ and $CO_2$ through snow during the winter season is an important feature of the annual flux cycle. It indicates a less than complete ice cover over the fen unlike the ice cover on the lake. This limited the trapping and the buildup of $CH_4$ in the fen during winter. A previous study showed that $CH_4$ emissions from the fen at ice-out are correlated with air temperature thus with daily snow melt and release of trapped gases (Jammet et al., 2015) as seen elsewhere (Friborg et al., 1997; Gažovič et

al., 2010). The flux rates during the ice-out season were however much lower than during the summer. Both $CH_4$ and $CO_2$ fluxes from the lake peaked during the ice-out season. These pulses coincided with the time of complete water overturn following lake ice thaw, (Fig. 2) and can be explained by the release of gases previously stored in and under lake ice. The annual seasonality was measured from 2.5 years of measurements, and further years of observations are needed to evaluate and explain inter-annual variability in the magnitude of the emissions.

The mean annual $CH_4$ efflux from the lake is in line with a regional estimate of ebullition flux in post-glacial and glacial lakes (32.2 mg $CH_4$ m$^{-2}$ d$^{-1}$, Wik et al. (2016)). Mean fen $CH_4$ fluxes agreed well with chamber measurements conducted over the same period in *Eriophorum*-dominated plots in the Stordalen Mire (P. Crill, unpublished data). The mean $CH_4$ flux measured during the ice-free seasons is within the range of summer $CH_4$ fluxes measured in northern wetlands that are dominated by sedges (Olefeldt et al., 2013). Measured $CO_2$ flux rates at the fen site also agreed with previous EC studies

within the wettest part of the mire (e.g. Christensen et al., 2012) and with flux rates measured with gas chambers in sedge-dominated vegetation plots in previous years (Bäckstrand et al., 2010) and during the study period (P. Crill, unpublished data), at the seasonal and annual scales. The fen is thus representative of minerotrophic northern fens where high $CH_4$ emissions are measured, due to year-round anoxia in the soil and to the dominance of vascular plants (Olefeldt et al., 2013).

Lakes that freeze solid in winter are not expected to emit a significant amount of $CO_2$ at the surface, unless ice-free holes

caused by strong bubble seeps are present (e.g. Sepulveda-Jauregui et al., 2015), which we do not observe in the lakes of Stordalen (Wik et al., 2011). The observation of $CO_2$ fluxes from the lake during the second winter at rates that are within the magnitude of land winter respiration (Fig. 3) was thus unexpected. The high winter flux rates were coincident with strong winds, increasing air temperature and high latent heat flux. Whether these are due to a physical evasion of $CO_2$ through snow over the lake surface or to lateral advection of land-emitted $CO_2$ is unclear. The extended flux footprint in winter, which

includes lake shores, might include part of the land in the middle of the lake, leading to vegetation-like flux magnitudes. Although this effect was limited by filtering for large wind dispersion during winter periods, part of the flux could still be influenced by land respiration. Furthermore, the impact of self-heating on the open path $CO_2$ flux measurements (Burba et al., 2008) is a potential issue, especially for such low range fluxes on the lake side, but could not be correctly quantified in this study and was thus not applied to avoid a potentially large systematic error. Besides, the self-heating issue of this gas

analyzer model results usually in an apparent uptake rather than in an excessive degassing at very cold temperatures (Burba et al., 2008).



During the summer, the small flux range resulted in a low signal-to-noise ratio (high relative random error). The uncertainty linked to the inclusion of low-frequency contributions is a problem that has been discussed widely in the eddy covariance community, particularly for low-flux environments (e.g. Sievers et al., 2015). The lake studied here can be qualified as a low-flux environment with respect to $CO_2$ exchange during the open water season. Testing $CO_2$ flux calculation with a new

method that removes the low frequency contributions (Sievers et al., 2015) on a portion of our data in July 2012 showed nevertheless that the summer fluxes at the lake were coincident with our measurement, showing a negative mean and diel pattern with slight uptake during the day (J. Sievers, personal communication, 2016). This confirms that despite a high noise our $CO_2$ lake flux measurements during summer are trustworthy. Besides, the flux footprint was representative of each ecosystem (lake vs. fen). The diel patterns of sensible and latent heat fluxes from the lake resembled those observed in boreal

lakes (Mammarella et al., 2015; Nordbo et al., 2011; Shao et al., 2015; Vesala et al., 2006), which further supports that our EC measurement from the eastern sector are representative of the lake surface.

## 4.2 Season-dependent transport pathways of $CH_4$ and $CO_2$ from the lake to the atmosphere

### 4.2.1 Ice-free season

Eddy covariance measures a direct flux across the surface-atmosphere interface, spatially integrating over m$^2$ to km$^2$ all

emitting pathways that are responsible for the transport of gas from the ecosystem to the atmosphere (i.e., total flux). In this study, $CH_4$ emissions from the lake during the ice-free seasons were dominated by short, large degassing events (Fig. 3, Fig. C1) that coincided with drops in atmospheric pressure. Furthermore, daily EC observations coincided with spatially averaged ebullition fluxes measured with bubble traps (Fig. 6). These observations suggest that the total $CH_4$ efflux from the lake during the ice-free seasons was mostly due to the release of bubbles formed in the sediments (ebullition), in line with

previous observations that ebullition is the main pathway for $CH_4$ emissions in shallow lake areas (Bastviken et al., 2004).

A strong relationship was found between $CH_4$ efflux and surface sediment temperature, thus the slight seasonality in summer $CH_4$ emissions from the lake (Fig. 5) is likely due to the seasonal increase in temperature in the production zone. This seasonal trend also supports a bubble release mechanism, since a seasonal increase in sediment temperature favors methanogenesis but also a decrease of $CH_4$ solubility (Casper et al., 2000; Wik et al., 2013). This means that the amount of

$CH_4$ emitted at the lake surface is directly linked to the amount of $CH_4$ produced within the sediments, as it has been observed using bubble-traps (Wik et al., 2014). A significant relationship between bubble flux and surface sediment temperature similar to the one we reported here, was observed in the lakes of the Stordalen catchment by Wik et al. (Wik et al., 2014), who identified the threshold for ebullition in the Stordalen lakes at 6˚C. Our EC system does measure fluxes for sediment temperatures under 6˚C, which could be diffusive. The occasional occurrence of degassing during summer that

timed up with short de-stratification events (Fig. S3) indicates that hydrodynamic transport and diffusion of $CH_4$ linked to




lake mixing may happen in this lake, as it has been observed in a boreal lake (Podgrajsek et al., 2014). Water currents can also trigger bubble release by disturbing surface sediments (Joyce and Jewell, 2003).

Exchange of $CO_2$ across the lake-air interface is mainly diffusion-limited due to the temperature of dissolution of $CO_2$ in water, which does not favor the release of $CO_2$ in bubbles (Tranvik et al., 2009). On average, the net $CO_2$ flux at the surface

of the lake during the ice-free seasons revealed photosynthetic activity. The strong light response curve of median diel emissions (Fig. 9) is largely influenced by flux rates measured during the warm, sunny summer of 2014. The presence of the diel pattern in summertime $CO_2$ fluxes was tested for the influence of advection from the lake shore by recalculating the fluxes from the lake using a 5 min average instead of 30 min (Eugster, 2003; Podgrajsek et al., 2015; Vesala et al., 2006). The pattern persisted for 5 min averaged fluxes during the summer of 2012 (Fig. S5), which suggests that advection had a

small effect on the summer $CO_2$ fluxes.

In an Alaskan lake (Eugster, 2003), negative flux of $CO_2$ from the atmosphere to the lake surface was attributed to advection because the fluxes occurred during stable atmospheric condition and negative sensible heat flux (H) thus a warming of the lake surface. We observed an opposite relationship here; $CO_2$ uptake at the lake in summer was associated with unstable atmospheric condition and positive H (Table 2). The diel pattern in H flux from the lake is coherent with the observation of

surface water temperature being systematically higher than air temperature during morning hours (not shown), resulting in the cooling of the lake surface. It was in antiphase with the diel patter in $CO_2$ fluxes (Fig. 8). Water-side convection due to cooling of the lake surface has been shown to enhance the diffusion-limited exchange of $CO_2$ (Eugster, 2003; Podgrajsek et al., 2015). If the lake water is under-saturated in $CO_2$ with respect to the atmosphere, this results in a downward $CO_2$ flux. Diel pattern in $CO_2$ flux linked to lake mixing have been observed in other eddy covariance studies, where it was associated

with a release of $CO_2$ to the atmosphere (Mammarella et al., 2015; Podgrajsek et al., 2015).

### 4.2.2 Ice-out season

The strong correlation of $CH_4$ and $CO_2$ emissions at the lake during both ice-out periods suggests that the gases were emitted to the atmosphere via the same mechanism, i.e. by turbulence-driven release of gases that have been accumulating in the lake. Conversely, the correlation is very low during the ice-free periods, when ebullition is the dominant process of $CH_4$

release. The outgassing pattern at ice-out coincided both years with the breakdown of thermal stratification in the water column after complete ice disappearance. Bubbles trapped in the winter ice of lake Villasjön contain both $CH_4$ and $CO_2$ (Boereboom et al., 2012). $CO_2$ stored in lake ice during winter can originate from benthic respiration which can occur under ice while dead plants from the previous summer are decomposing (Karlsson et al., 2008). Methanotrophy can be important during overturn events in lakes (Kankaala et al., 2006; Schubert et al., 2012) thus $CO_2$ may also be produced in

lake water during ice thaw, which lasts several days, as an output of $CH_4$ oxidation. Dissolved $CO_2$ could also enter the lake as catchment input via lateral meltwater run-off before complete overturn (Denfeld et al., 2015).




The processes underlying $CH_4$ degassing during ice-out in 2013 have been discussed in details in a previous study (Jammet et al., 2015) and it is likely that $CO_2$ was release via the same physical mechanisms. The degassing pattern observed in 2013 was repeated in 2014, with a mean and median $CH_4$ flux rate smaller than the previous year, but still significantly higher than the $CH_4$ emissions of the ice-free season (Fig. S2). In 2014 the ice-out period started earlier but was longer (Figure 2), and our measurement system may have missed part of the degassing due to instrument failure. As ice thaws, gases contained in bubbles can migrate to the water (Greene et al., 2014) and be released to the atmosphere when thermal stratification gradually breaks because of the warming up of the water column. We can speculate that a delay in the timing of overturn following ice thaw may favor oxidation of $CH_4$ within the water column when it is already partly mixing, which would raise the concentration of dissolved $CO_2$ in the water and could contribute to a smaller burst of $CH_4$ during complete overturn.

## 4.3 Annual atmospheric carbon budget

### 4.3.1 Carbon function

The fen was an annual sink of carbon with respect to the atmosphere, while the lake was an annual source, at a magnitude representing 70 % of the fen sink. The total annual C-emission from the lake is within the range of annual C-emissions ($CH_4+CO_2$) from lakes of subarctic Sweden (5 to 54 g C m$^{-2}$ yr$^{-1}$, Lundin et al., 2015) estimated mostly using water grab sampling.

At the fen, we report a stronger summer sink of $CO_2$ (three ice-free seasons average -206.8 g C m$^{-2}$, Table 3) compared to earlier studies in the inner fens of the Stordalen mire (-133 g C m$^{-2}$, years 2001-2008, Christensen et al. 2012), but annually a net $CO_2$ uptake (-58.5 to -79.1, average -66.3 g C m$^{-2}$ yr$^{-1}$) that is similar to the 2001-2008 average (-66 g C-$CO_2$ m$^{-2}$ yr$^{-1}$, Christensen et al 2012) and smaller than the average for years 2006-2008, which were warm years (-90 g C-$CO_2$ m$^{-2}$ yr$^{-1}$, Christensen et al. 2012). The difference is due to the higher $CO_2$ respiration we measured in winter, which equaled 54 % of the summer sink on average during the measuring period. The annual $CH_4$ emissions of 21.2 C-$CH_4$ m$^{-2}$ yr$^{-1}$ (Table 3) is very close to what has been reported for the internal fens of Stordalen in an eddy covariance study where winter emissions were estimated with a temperature relationship (Jackowicz-Korczyński et al., 2010). This highlights the stability of the fen in term of $CH_4$ emissions but also the low sensitivity of the annual sum to the choice of gap filling method for the fen $CH_4$ flux dataset, which is tightly linked to temperature.

To determine whether an ecosystem is a net source or sink of carbon within the landscape carbon cycling, a full net ecosystem carbon balance (NECB) must take into account both vertical carbon exchange but also lateral carbon transport, in and out of the system (Chapin et al., 2006). In 2008, net DOC export at the fen was 8.1 g C m$^{-2}$ yr$^{-1}$ and net POC export was 0.6 g C m$^{-2}$ yr$^{-1}$ (Olefeldt and Roulet, 2012). Combined with our annual atmospheric carbon budget (Table 3), this results in a fen NECB of -29.5 g C m$^{-2}$ yr$^{-1}$ in the first year and -43.3 g C m$^{-2}$ yr$^{-1}$ in the second year. These numbers are marginally smaller than the long-term carbon accumulation of -51 g C m$^{-2}$ yr$^{-1}$ inferred from the analysis of a peat cored in Stordalen



and attributed to a period when the mire was dominated by graminoids (Kokfelt et al., 2010). We are not aware of existing data on net export of DOC and POC through the lake to make a similar estimate.

In term of radiative forcing, considering the 28-fold stronger global warming potential of atmospheric $CH_4$ vs. atmospheric $CO_2$ over 100 years (GWP100, Myhre et al. 2013), vertical carbon exchange has a warming impact on the atmosphere at both ecosystems through their net annual emissions of $CH_4$. Annual estimates that disregard winter and transitional seasons are likely missing part of the annual carbon emissions from seasonally freezing lakes and wetlands.

### 4.3.2 The lake as a summer $CO_2$ sink

Because of dynamic external and internal factors governing the consumption and production of $CO_2$ in surface waters, the $CO_2$ function of a lake can vary seasonally (Maberly, 1996; Shao et al., 2015). Lake Villasjön was an annual source of $CO_2$ due to the spring outgassing, but it was a small sink of $CO_2$ in the open-water period. While flux rates in summer 2012 and 2013 were negative but close to the noise level, the uptake was larger and significant in 2014 when the summer was hotter and sunnier. Averaged estimates from water sampling measurements in the lakes of the Abisko area indicate the lakes to be mainly $CO_2$ sources during the summer, except for a few lakes that were seasonal $CO_2$ sinks during the ice-free season (Karlsson et al., 2013). In the few eddy covariance studies available from Arctic and boreal sites, lakes are reported as $CO_2$ sources during the ice-free season (Lohila et al., 2015; Mammarella et al., 2015; Podgrajsek et al., 2015) and occasional $CO_2$ sinks during the warm summer months, while being sources on the seasonal scale (Anderson et al., 1999; Eugster, 2003; Huotari et al., 2011; Jonsson et al., 2008).

Although Villasjön is representative of a widespread postglacial lake type across subarctic and Arctic latitudes, it differs from most lakes studied in the northern lakes literature due to its particularly shallow depth, which results in the lack of long-term stratification during the open-water season. Lakes that are similarly shallow are often thermokarst lakes or peatland ponds (Vonk et al., 2015). Summer $CO_2$ uptake at the level of what we report here has been observed in highly productive lakes (Pacheco et al., 2013) or in thaw ponds colonized by submerged plants and microbial mats (Laurion et al., 2010; Tank et al., 2009). Estimates of air-lake carbon exchange using water sampling and floating chambers (Karlsson et al., 2013) showed that a minority of lakes in subarctic Sweden were $CO_2$ sinks in summer with a total seasonal $CO_2$ exchange from -3.8 to -10 g C m$^{-2}$ yr$^{-1}$, while being large sources at ice-out, offsetting the summer sink.

Lakes having poor hydrological connections with their upstream catchment have been reported in previous studies to be net $CO_2$ sink in summer, e.g. in Minnesota (Striegl and Michmerhuizen, 1998) or in thaw ponds of the Canadian Arctic (Tank et al., 2009). In the latter study, within-lake DOC was proposed to occur as a byproduct of macrophyte photosynthesis, showing that net $CO_2$ uptake in lakes is not always associated with low DOC concentrations. In large and shallow lakes surrounded by peatlands, vegetation develops on the sediment surface thanks to the presence of humic acids supplied by the



peaty shores and a well-illuminated bottom (Banaś et al., 2012). The analysis of peat and lake sediment records in Stordalen suggested that a significant amount of peat is exported from the mire to lake Villasjön during periods of mire erosion, likely due to permafrost thaw (Kokfelt et al., 2010). This lake may have high nutrient content due to peat input at the shores, organic-rich sediments and autochtonous vegetation. Lakes that do not stratify tend to be more productive because of the

more regular mixing of nutrients (Tranvik et al., 2009; Wetzel, 2001).

### 4.3.3 Influence of overturn on annual C-emissions

On an annual scale, the ice-out period accounted for 50 % of annual carbon exchange ($CH_4 + CO_2$) at lake Villasjön and turned the lake from a summer $CO_2$ sink to an annual source. In other, deeper lakes that stratify in summer and don't fully mix in spring, fall overturn led to the highest emissions of $CH_4$ or $CO_2$ of the year (Kankaala et al., 2006; Schubert et al.,

2012), accounting for a large part of the annual $CO_2$ flux (Huotari et al., 2011). Other seasonally ice-covered lakes emitted large amounts of $CH_4$ and $CO_2$ following ice-out (Anderson et al. 1999, Karlsson et al. 2013), while high concentrations of dissolved $CO_2$ and $CH_4$ under lake ice has been measured in Northern American lakes (Striegl and Michmerhuizen, 1998), across lakes of the Swedish subarctic (Karlsson et al., 2013), Alaskan thermokarst lakes (Sepulveda-Jauregui et al., 2015) or in thaw ponds in Canada (Tank et al. 2009). Accumulation of $CH_4$ and $CO_2$ under the ice is thus a general feature of lakes

with an anoxic hypolimnion or sediment in winter, but studies reporting direct measurement of the outgassing of $CH_4$ and $CO_2$ at the lake surface right after ice-out are scarce because it is a rapid and variable phenomenon that is seldom included in direct flux measurements.

The large impact of $CO_2$ and $CH_4$ release during spring has been observed in lakes of the Abisko area where water samples before and after ice-out were used to estimate the thaw release, which accounted on average for 45 % of annual emissions

(Karlsson et al., 2013; Lundin et al., 2013). A few regional studies reported on a lesser importance of the spring season on annual carbon emissions from lakes. Sepulveda-Jauregui et al. (2015) sampled 40 thermokarst Alaskan lakes where the maximum emission of $CH_4$ and $CO_2$ were measured in summer. Many of these lakes were thermokarst lakes that continuously emitted $CH_4$ in winter through open holes, which we don't observe in lake Villasjön. Thermokarst lakes are usually stronger $CH_4$ emitters than post-glacial lakes on a per unit area basis, yet post-glacial lakes seem to be a larger

overall source because they cover a larger area in the high northern latitudes (Wik et al., 2016). A recent review by Wik et al. (2016) compiled $CH_4$ emissions from several types of lakes. Over the total of 733 sites, ice-out was estimated to contribute ~23 % of annual emissions of lakes and ponds. Only four sites were measured with EC and those four only comprised ice-free season (July–August) measurements. Thus the comparison to our results is limited by differences in temporal and spatial scale of the methods.

Our study underlines the high significance of shoulder seasons (more precisely, overturn periods following periods of gas storage) for the biogeochemistry of lakes and the emission of $CO_2$ and $CH_4$ to the atmosphere. The relative importance of

these periods on the annual emissions depends on the extent of the overturn (Huotari et al., 2011; Kankaala et al., 2006), the extent of methanotrophy during and before full lake mixing (Kankaala et al., 2006; Schubert et al., 2012), as well as the amount of degradable organic matter in the hypolimnion. Although these overturn periods cover only a few weeks or days, they are important for both $CH_4$ and $CO_2$ emissions in lakes and should be included in measurement campaigns when feasible.

## 5.     Conclusions

The waterlogged fen and the shallow lake showed contrasting annual cycles in term of $CH_4$ and $CO_2$ exchange with the atmosphere. This difference is explained, first, by the presence of an ice lid over the lake surface which led to the storage of gases in winter and large subsequent emissions in spring, while evasion of $CH_4$ and $CO_2$ to the atmosphere from the fen in the wintertime limits the importance of emissions during ice and snow melt. Second, the dense cover of vascular plants at the fen leads to high $CH_4$ emissions and $CO_2$ uptake in summer.

Annually, the fen was a net carbon sink with respect to the atmosphere, while the lake was a source of carbon due to the degassing in spring that outweighed the uptake of $CO_2$ in summer. This study confirms the importance of overturn period in lakes for both $CH_4$ and $CO_2$ annual emissions. The magnitude of the degassing at ice-out may depend greatly on lake type, morphometry and productivity status. The lake studied represents a common type of shallow postglacial lake across the subarctic latitudes. Further direct measurements of surface fluxes covering several years and different lake types are needed to evaluate the inter-annual variability in the magnitude of the degassing in shoulder seasons as well as its importance for the annual emissions of northern lakes in general.

Finally, ebullition was identified as the main transport pathway for $CH_4$ emissions in the shallow subarctic lake and a net $CO_2$ sink in summer indicated large photosynthetic activity. Turbulence–driven diffusive release of $CO_2$ and $CH_4$ was predominant during spring overturn following ice-out. These results show the potential of the EC method in lake environments for a better understanding of flux processes and annual seasonality in the understudied but abundant postglacial lakes and ponds.

### Data availability

The eddy covariance data and meteorological data used in this study are available upon request to the lead author. A version of the flux dataset before flux source partitioning between lake and fen is available on the FLUXNET database with ancillary data.



**Authors contributions**

TF, MJ, PC designed the study. MJ collected, analyzed and interpreted the data. SD developed and performed the gap filling modeling. EK pre-processed part of the eddy covariance raw data. FJWP performed the 2D footprint modeling and drew the footprint figure. MW provided methane ebullition data. MJ wrote the manuscript, figures and all authors commented on it.

**Acknowledgements**

This work was funded through the Nordic Centre of Excellence, DEFROST, under the Nordic Top-Level Research Initiative, and the collaborative research project Changing Permafrost in the Arctic and its Global Effects in the 21st century (PAGE21). We thank the EU-funded International Network for Terrestrial Research and Monitoring in the Arctic (INTERACT) for financing visits at the field station, the Danish National Research Foundation for supporting activities within the Center of Permafrost (CENPERM, DNRF100) and the Abisko Scientific Research Station for providing field work infrastructure. We thank Tyler Logan, Fabian Rey, Robert Holden, Niklas Rakos and Mathias Madsen for technical assistance and maintenance on the field.

**Appendices**

15 **Appendix A: Details in eddy covariance flux calculation**

Processing of the raw eddy covariance data for flux calculation included despiking (Vickers and Mahrt, 1997), angle of attack correction on raw wind components (Nakai et al., 2006), 2-D axis-rotation correction (Wilczak et al., 2001) on wind speed components, and detrending of 30 min. raw data intervals by block averaging the vertical wind speed and scalar signals (Moncrieff et al., 2004). The time delay between vertical wind speed and each scalar ($CO_2$, $CH_4$, $H_2O$, sonic

20 temperature) was removed by finding the maximum of the cross covariance function of vertical wind speed and each scalar (Fan et al., 1990). The time-window search was adjusted for each gas and each period when a change in the set-up occurred.

The effect of density fluctuations on $CO_2$ fluxes was corrected (Webb et al., 1980). The correction lowered the amplitude of the $CO_2$ flux dataset on average by 53% (slope of the linear regression between non-density corrected $CO_2$ fluxes and final corrected fluxes = 0.47, $r^2 = 0.73$). For $CH_4$ fluxes, density fluctuations were compensated after flux calculation using

25 the formulation of Ibrom et al. (2007a), including the pressure-induced fluctuation term. Comparing $CH_4$ fluxes with and without density correction showed a difference of less than 5% in the flux magnitude. Turbulent fluxes calculated with the eddy covariance method are affected by spectral losses due to the instrumental setup and the limited time response of the instruments. Losses in the low frequency range due to the finite flux averaging time were corrected analytically after Moncrieff et al. (2004). $CO_2$ flux loss in the high frequency range was also corrected analytically (Moncrieff et al., 1997),

while $CH_4$ fluxes derived from the closed-path system required an *in situ* assessment of the system's cut-off frequency (Ibrom et al., 2007b) due to the long sampling line. This assessment was done separately for each period with a continuous instrumental set-up and the associated flux attenuation was calculated and compensated following the formulation by Horst (1997). The magnitude of the spectral loss hence of the total spectral correction was on average 31% for $CO_2$ fluxes and
37% for $CH_4$ fluxes.

**Appendix B: Performance of the ANN models**

The ANN gap filling method was most performant on the fen dataset, achieving an $r^2$ of 0.88 during the training phase, and an $r^2$ of 0.85 between measured and predicted values over the whole dataset (expressing the capacity of generalization of the
model) with a relative mean square error (RMSE) of 23 % (Figure B1). The ANN gap filling of lake $CH_4$ fluxes achieved an $r^2$ of 0.71 in the training phase and an $r^2$ of 0.70 between predicted and measured fluxes on the whole dataset, with an RMSE of 51 %. The lake model was most accurate for periods with the best data coverage in the measured dataset (spring seasons, $r^2 = 0.77$). The lower accuracy of the model during the ice-free seasons ($r^2 = 0.47$) is also due to the pulse-character of lake ebullition, which was not always reproduced by the model, while the background seasonal trend was present.

For the fen $CH_4$ fluxes, the model was most accurate during the ice-free seasons when fen $CH_4$ emissions are tightly linked to peat temperature and least performant during the ice-out periods. Unsurprisingly, the prediction performance of the models was dependent on data coverage in the original dataset. On the annual scale, both fen and lake models were most performant during the first year (June–May), which had the least amount of data loss, with an $r^2$ of 0.88 (RMSE 23 %) on the first year for the fen model and of 0.82 (RMSE 40 %) for the lake model between predicted and measured fluxes.

The ANN modeling was likewise performant on the fen $CO_2$ flux dataset, achieving an $r^2$ of 0.86 (RMSE 35 %) over the whole dataset between measured and predicted value (Figure B1). The model was most performant during the ice-free seasons. This can be explained by a better data coverage but also by better constrained processes during the growing season and on the annual scale, than during the snow-cover season, when additional drivers than the one selected as model inputs may play a role in the variability of the fluxes at the half-hourly scale (sensitivity to footprint changes, turbulence, variation
in snow depth, partial melt, etc.).

**Appendix C**

Figure C1.



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




**Tables**

**Table 1**: Climatic conditions per season: air temperature measured at the Abisko Scientific Research Station, peat temperature at 10 cm depth in the fen, surface sediment temperature in the lake (1 m depth), total net radiation at the fen surface and at the lake surface. Season delimitations are: 1 June 2012 – 14 Oct 2012 ; 26 May 2013 – 15 Oct 2013 ; 2 June 2014 – 9 Oct 2014 ; 15 Oct 2012 – 14 Apr 2013 ; 16 Oct 2013 – 10 Apr 2014 ; 15 Apr 2013 – 25 May 2013 ; 11 Apr 2014 – 1 June 2014 ; 1 June 2012 – 31 May 2013 ; 1 June 2013 – 31 May 2014 (see text for a definition).

| Season | Year | Length (days) | Air T (°C) | Peat T at 10 cm (°C) | Surf. sed. T (°C) | Total Rn, fen ($10^3$ $Wm^{-2}$) | Total Rn, lake ($10^3$ $Wm^{-2}$) |
|---|---|---|---|---|---|---|---|
| Ice-free | 2012 | 136 | 7.8 | 8.9 | 10.2 | 375 | 477 |
| | 2013 | 143 | 10.2 | 9.9 | 12.0 | 467 | 555 |
| | 2014 | 130 | 10.4 | 10.5 | 12.8 | 545 | 609 |
| Ice-cover | 2012-2013 | 182 | -7.9 | 0.0 | 0.7 | -168 | -196 |
| | 2013-2014 | 177 | -6.5 | 0.2 | 0.8 | -151 | -207 |
| Ice-out | 2013 | 41 | 4.3 | 0.3 | 1.9 | 112 | 115 |
| | 2014 | 52 | 2.2 | 1 | 1.9 | 117 | 248 |
| Annual | 2012-2013 | 365 | -0.3 | 3.5 | 4.7 | 374 | 461.6 |
| | 2013-2014 | 365 | 0.9 | 4 | 5.1 | 377.2 | 528.5 |



**Table 2**: Spearman's rank correlation coefficient ($\rho$) showing the degree of association between half-hourly lake $CH_4$ flux and lake $CO_2$ flux with potential drivers of variability. All data were grouped per season. See Table 2 for the limitation of the seasons. Lake fluxes are filtered for high lateral wind speed in winter.

| | Lake $CH_4$ flux (Spearman's $\rho$) | | | | Lake $CO_2$ flux (Spearman's $\rho$) | | | |
|---|---|---|---|---|---|---|---|---|
| | **Annual** | **Ice free** | **Ice-out** | **Ice cover** | **Annual** | **Ice free** | **Ice-out** | **Ice cover** |
| **Air T** | 0.36*** | 0.40*** | 0.26*** | 0.14** | -0.21*** | -0.22*** | 0.38*** | 0.16*** |
| **T water surface** | 0.27*** | 0.48*** | 0.37*** | 0.18* | -0.21*** | -0.21*** | 0.27*** | 0.09*** |
| **T bottom** | 0.24*** | 0.49*** | 0.49*** | 0.10* | -0.26*** | -0.21*** | 0.40*** | -0.10*** |
| **Wind speed** | 0.36*** | 0.32*** | 0.40*** | 0.27*** | 0.26*** | 0.13*** | 0.38*** | 0.36*** |
| **H flux** | -0.15*** | 0.12*** | -0.18*** | -0.12* | -0.62*** | -0.67*** | -0.38*** | -0.50*** |
| **LE flux** | 0.24*** | 0.38*** | 0.16*** | 0.08 | -0.29*** | -0.49*** | 0.05 | -0.14*** |
| **Solar radiation** | 0.27*** | 0.13*** | -0.02 | -0.11* | -0.09*** | -0.52*** | 0.02 | -0.22*** |
| **$CO_2$ flux** | 0.41*** | -0.13*** | 0.67*** | 0.21** | -- | -- | -- | -- |

*** p-value < 0.001;  ** p-value < 0.01; * p-value < 0.1





**Table 3:** Seasonal and annual sums of $CO_2$ and $CH_4$ fluxes after gap-filling, in g C $m^{-2}$ $y^{-1}$. Sum BE is the bias due to the gap filling model (Eq. (1)), scaled to annual flux units and multiplied by the number of gaps in the flux dataset over the season or year. Sum of $CO_2$ fluxes from the lake is only reported for the first year.

| Season | Year | Fen $CH_4$ flux | | Lake $CH_4$ flux | | Fen $CO_2$ flux | | Lake $CO_2$ flux | |
|---|---|---|---|---|---|---|---|---|---|
| | | Total flux | Sum BE | Total flux | Sum BE | Total flux | Sum BE | Total flux | Sum BE |
| Ice-free | 2012 | 14.8 | 0.1 | 2.5 | -0.1 | -179.4 | -14.8 | -24.4 | <0.001 |
| | 2013 | 16.2 | -0.1 | 2.8 | -0.1 | -201.3 | 13.2 | nc[*] | nc |
| | 2014 | 16.8 | 0.09 | 3.0 | -0.5 | -239.6 | -5.1 | nc | nc |
| Ice-cover | 2012-2013 | 4.1 | -0.2 | 0.3 | -0.3 | 109.6 | -8.8 | 12.4 | <0.001 |
| | 2013-2014 | 4.2 | 0.7 | 0.4 | -0.4 | 115.5 | 20.7 | nc | nc |
| Ice-out | 2013 | 1.4 | -0.1 | 2.5 | -0.2 | 11.3 | -0.4 | 33.3 | <0.001 |
| | 2014 | 1.7 | -0.07 | 1.3 | 0.5 | 11.7 | 1.1 | nc | nc |
| Annual | 2012-2013 | 20.3 | 0.2 | 5.3 | -1.9 | -58.5 | -26.1 | 21.5 | <0.001 |
| | 2013-2014 | 22.1 | -0.1 | 4.4 | 0.6 | -74.1 | 97.4 | nc | nc |
| **Annual** | Average ± STD | 21.2 ± 1.3 | -0.2 | 4.9 ± 0.6 | -0.7 | -66.3 ± 11 | 35.7 | 21.5 | <0.001 |

[*]nc = not computed



# Figures

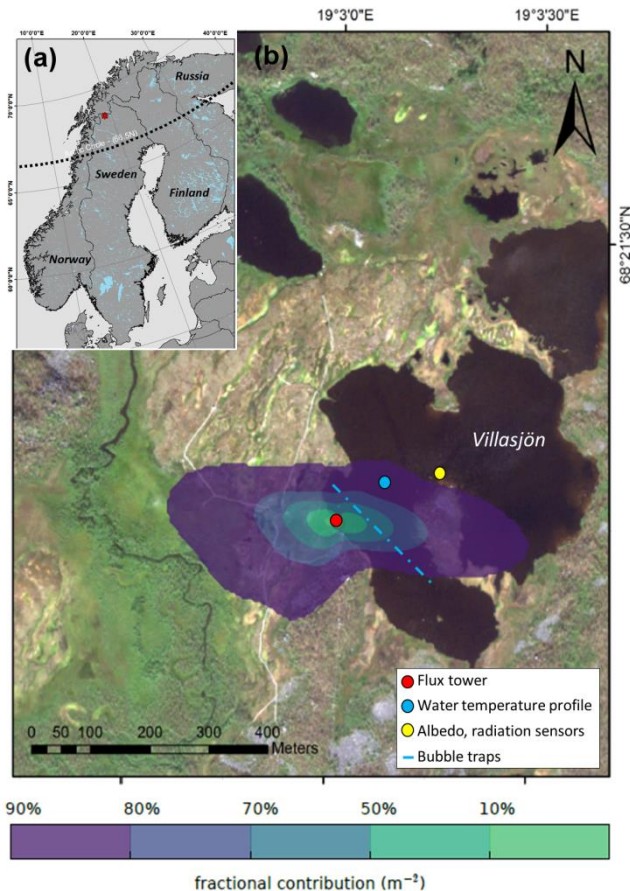

**Figure 1:** Location of the study site (a) and flux footprint of the flux tower in summer averaged over all years (b). The color scale indicates the extent of the fractional contribution from the source area to the fluxes measured at the tower. The location of the flux tower is indicated along with the location of the main environmental data sources.







**Figure 2:** Air, peat and surface sediment temperature (a) ; cumulative daily precipitation (b) ; net radiation input at the fen and the lake surfaces (c) ; albedo of the lake surface (d) ; temperature gradient in the lake water column defined as ($T_{w,10cm}$ - $T_{w,100cm}$) divided by the depth difference $\Delta z$ (e); water temperature profile in the lake center derived from continuous temperature measurements at depths 10 cm, 30 cm, 50 cm and 100 cm ( = bottom) (f).





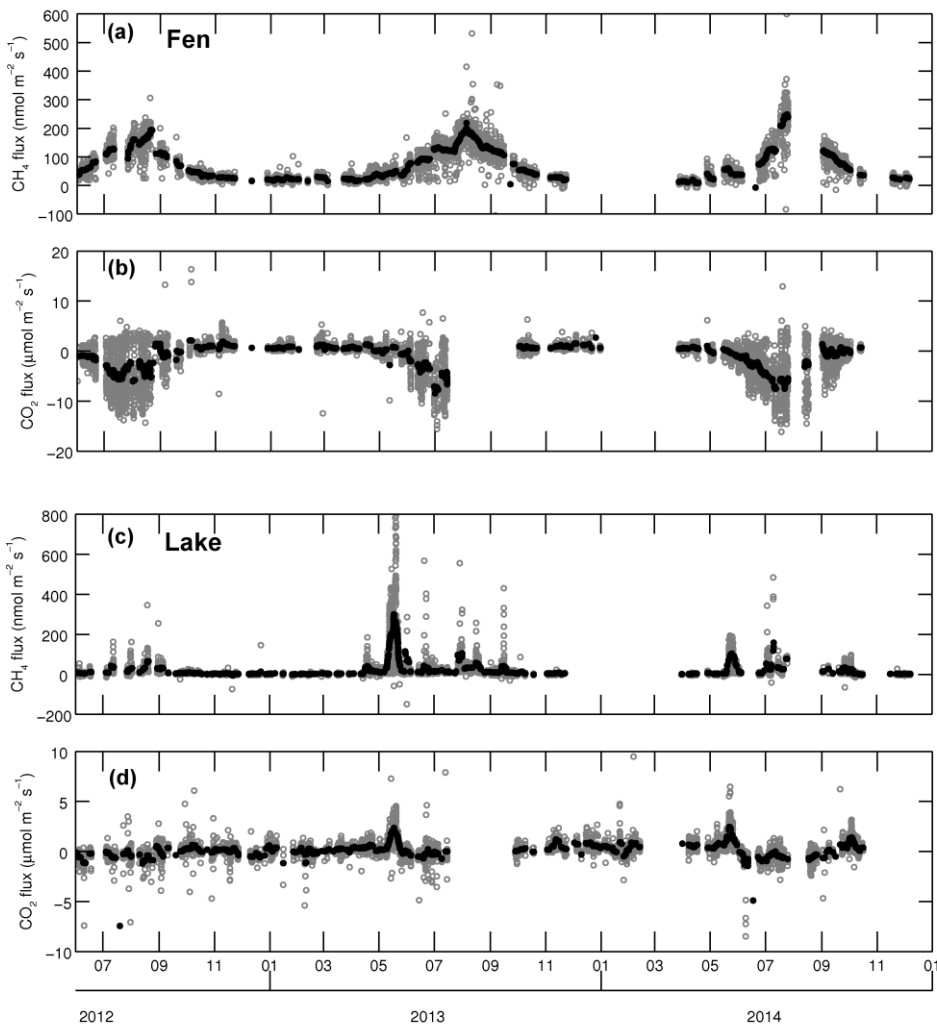

**Figure 3**: Measured $CH_4$ and $CO_2$ fluxes at the fen (a,b) and the lake (c,d) over the full study period. Light grey dots are half-hourly values and black dots show a 5-days running mean. Note on panel (c): 9 flux values above 800 nmol m$^{-2}$ s$^{-1}$ are not displayed for visibility.




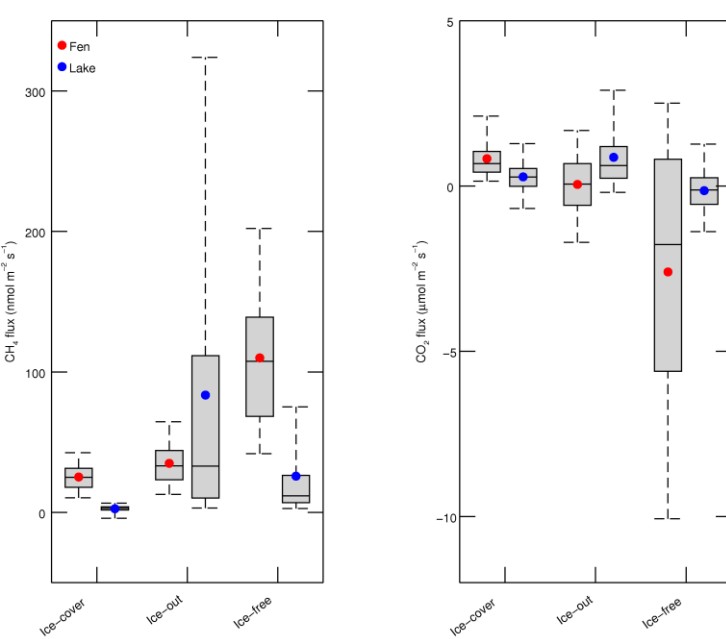

**Figure 4**: Measured flux rates of $CH_4$ (left panel) and $CO_2$ (right panel) per season and per ecosystem. The central line of the boxplots shows the median, box edges show $25^{th}$ and $75^{th}$ percentiles, and whiskers show $5^{th}$ and $95^{th}$ percentiles. The black dots indicate the mean flux rate. Outliers are not displayed.




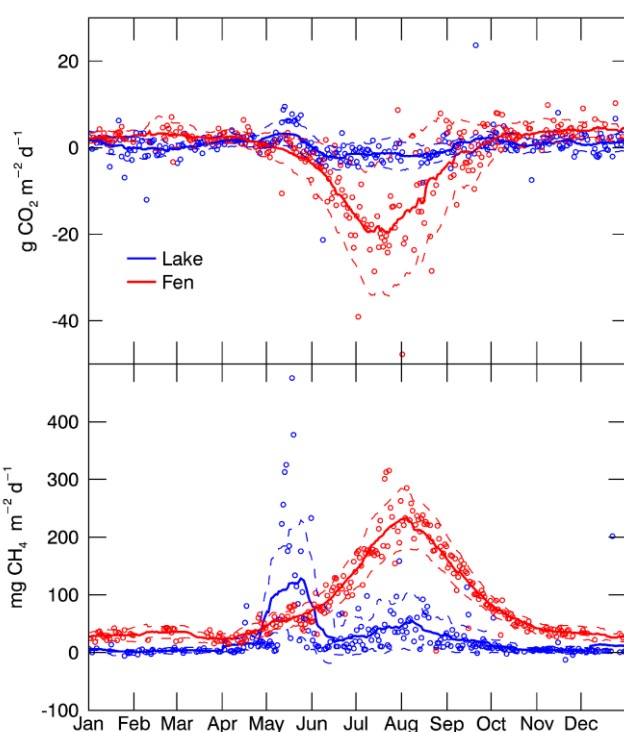

**Figure 5**: Averaged seasonality of $CO_2$ and $CH_4$ fluxes at both ecosystems. Dots show daily means across the whole measurement period, lines are a smoothing filter of the daily mean with a 30 days window. Dashed lines show the standard deviation around the daily means, smoothed with a 30 days window.




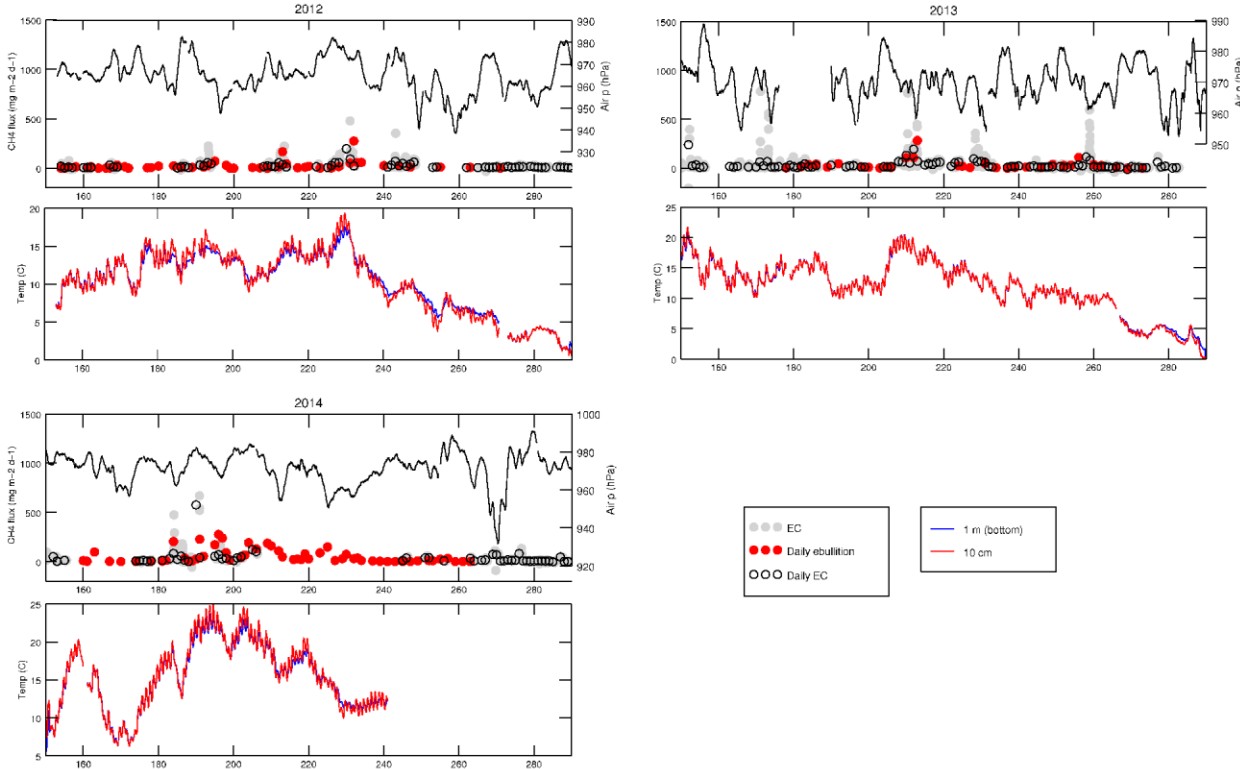

**Figure 6**: CH$_4$ emissions at the lake during the ice-free seasons of 2012, 2013 and 2014, from day 152 do day 290. Grey dots are half-hourly eddy covariance observations, open black dots are daily means of the eddy covariance fluxes and red dots show spatially-averaged daily CH$_4$ ebullition measured in the lake with bubble traps. Water temperature at depths 10 cm (red line) and 100 cm (sediment surface, blue line), as well as atmospheric pressure (black line) are also shown.



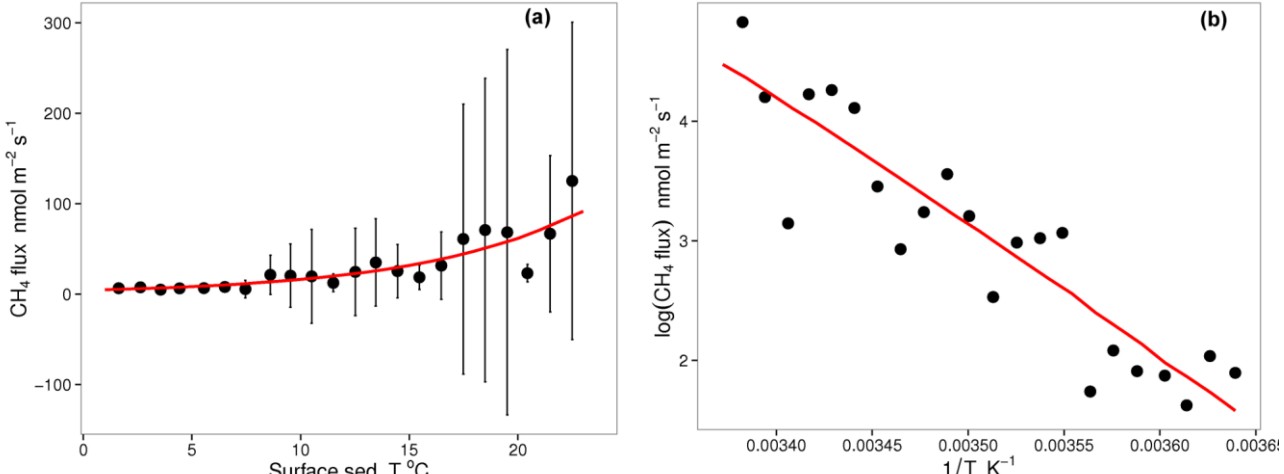

**Figure 7**: Relationship between lake $CH_4$ emissions and surface sediment temperature, using $CH_4$ flux rates averaged by bins of 1°C across the three ice-free seasons (a) and arrhenius plot of the natural logarithm of the flux versus the inverse surface sediment temperature in Kelvin (b). Error bars show the standard deviation of the $CH_4$ flux rates around the means within each averaging bin. The red lines are regression fits, with $r^2 = 0.82$ (*p<0.001*).




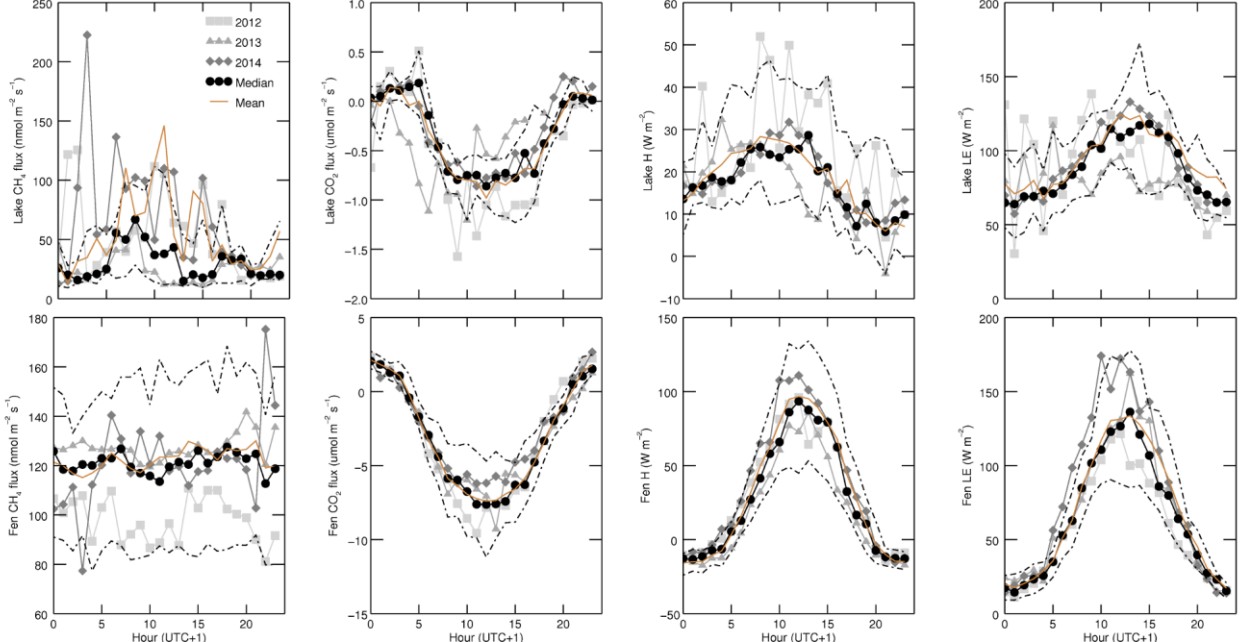

**Figure 8**: Diel medians, mean and $25^{th}$-$75^{th}$ percentiles of $CH_4$ fluxes, $CO_2$ fluxes, sensible heat flux (H) measured at the lake (upper panels) and at the fen (lower panels), from June to August of each year. Grey lines show the hourly median values for each year and the black line is a median of the three ice-free seasons.




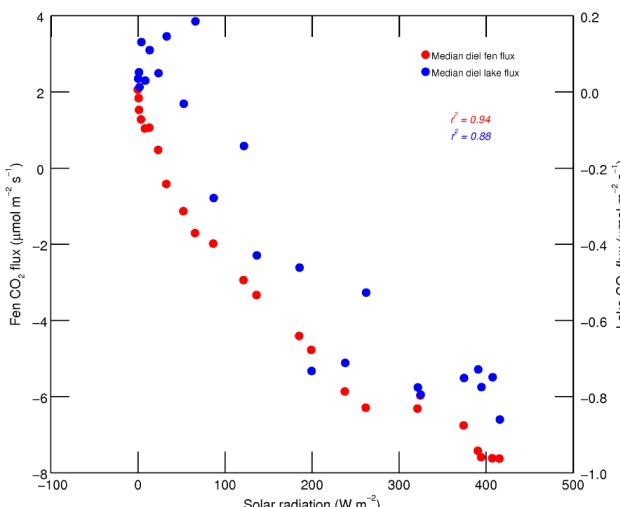

**Figure 9**: Diel median of the net $CO_2$ exchange at the fen (red) and at the lake (blue) versus diel median solar radiation, between June and August. Each dot is the hourly median flux of the combined three ice-free seasons. Note that the two y-axis have different scales.





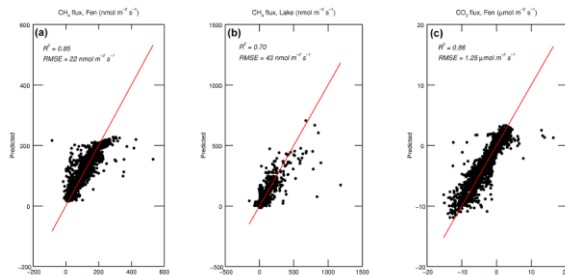

**Figure B1**: Evaluation of the artificial neural network models: measured vs. predicted hourly flux values for fen $CH_4$ fluxes (a), lake $CH_4$ fluxes (b) and fen $CO_2$ fluxes (c). In red is the 1:1 line.



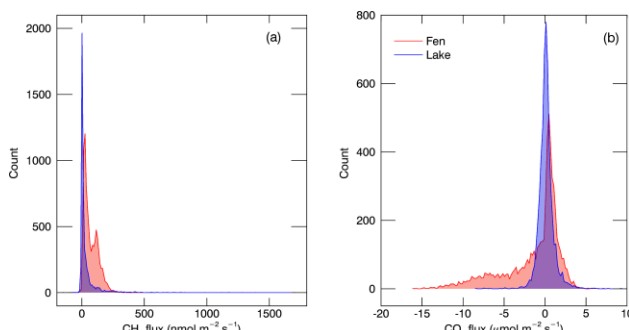

**Figure C1**: Probability density functions of (a) the measured $CH_4$ fluxes (bin size 10 nmol $m^{-2}$ $s^{-1}$) and (b) the measured $CO_2$ fluxes (bin size 0.2 µmol m-2 s-1) from the fen and from the lake during the entire measurement period.