# Peer review of "Year-round CH4 and CO2 flux dynamics in two contrasting freshwater ecosystems of the subarctic"

_Biogeosciences, 2016_

## Referee Comment (RC1) · I. Mammarella (Referee) · 6 Mar 2017

This manuscript presents two years of methane and carbon dioxide flux measurements at a lake and a fen sites. The dataset is very interesting, the framework analysis and results discussion very comprehensive and well written. I can recommend the final publication in Biogeosciences after the following comments are properly addressed:

1) Measurements of $CO_2$ fluxes were done by using an open path IRGA (LI-7500). Although this is a convenient or sometimes the only option in remote sites, I would use it with caution for measuring very low fluxes like the one presented here for the lake. The effect of air density fluctuations on $CO_2$ flux becomes very important, and the validity of WPL correction for low $CO_2$ flux has been also questioned in the past (see Ono et al., 2008).

2) I was a bit surprised to see that the high frequency corrections for $CO_2$ flux were so high (31% on average). Usually for open path EC system is much less. I guess this may be because of relatively large separation between the IRGA and the sonic anemometer. What was the separation? What is the value range of time lag for CO2 and which windows have You used? There may be problems with the WPL temperature term, which in theory should be measured in (or close) the path of the IRGA?

3) What would be the reason of relatively high (anti)-correlation between CO2 flux and H during wintertime (from Table 2)? And what about summer?

4) CH4 flux: it is not clear if H2O was measured by LGR. If not, then I guess the H2O fluxes measured by LI-7500 were used in the WPL correction for CH4. What EddyPro does when H2O comes from the LI-7500? The H2O fluctuations in the sampling cell of LGR may be quite different than the ones measured by the open path IRGA. How the authors cope with this issue? Why the compensation for the pressure term was also added? Which pressure data have been used for this?

5) Besides of density fluctuations caused by H2O, spectroscopic correction should be also applied to CH4 flux (see Peltola et al., 2014).

It seems to be some correlation between CH4 flux and LE (table 2). Is this because of points 4 and 5?

Summarizing, I know that there are limitations on including (in proper way) all these aspects, but at least it should be checked how important they are. Finally, the authors should acknowledge more clearly these aspects when discussing the uncertainty of these fluxes.

**Minor comments:**

Pag 6 L 11. Add also latent heat flux.

Pag6 L.18. "mixing ratio" usually means "dry mole fraction", but I guess this is "wet mole fraction", so without dilution correction. For calculating dry mole fraction point by point (high frequency data), simultaneous H2O measurements are needed.

Pag. L.26-28. How the synchronization was done? Just using the time stamps?

Pag.7 L.14. FST<0.3 is quite strict criteria. What about using FST<1? Is there a relevant difference in the data coverage?

Pag.7 L.15. Ustar threshold is taken equal to 0.1 m/s. How this was determined?

Pag.8 L8-10. Could you report some of these values used for the footprint calculation, e.g. roughness lengths?

Pag.8 L13-14. Is longer footprint in winter because of more stable conditions (negative H and low wind speed)? Or why?

Pag.8 L16. Based on what, the criterium $\sigma_v < 1$ m/s was used?

Pag.9L.25. Why the method did not perform well for lake $CO_2$ flux?

Pag10L12-13. The random error of fluxes is usually proportional to the flux magnitude. Do you mean the relative random error (error normalized by the flux) is smaller?

Pag.10 L21. How did you calculate the RE of the fluxes modeled with ANN?

Pag.11. L9. Is the thaw season the same as ice-out season?

Pag.12 L15-16. Do you mean Fig. 2e and Fig 2f ?

Pag. 13. L15. How do you explain this fall burst of $CO_2$?

Pag.15 L5-10. The diel cycle of 2012 H is quite noised respect to the other years. Why?

Pag.16 L10-20. May be some literature values can be added here for comparison.

Pag.16 L22-23. The highest correlation I can see for winter fluxes is with H.

Pag.16 L24. With EC it is not possible to measure advection, however You may see an increase of $CO_2$ mean concentration in the data, which may indicate non-turbulent transport of $CO_2$ from land.

Pag.16 L.27-31. I would say that it could be important to get a rough estimate of this correction. The effect (and direction) of this correction depends on the sign of H.

Pag.17 L.9-11. How the EBC (energy balance closure) plots look like in different years?

Pag.20 chapter 4.3.2. Are there any measurements of $pCO_2$ for this lake during summer? Or chamber measurements? Anything that could support the sink that you measured with EC?

Pag.23 L17. Coordinate rotation is not really a correction.

Pag.23 L19-22 . Please report the values of time windows and time lag. Was the lag maximization applied also for sonic temperature? The lag between w and Ts should be 0.

Pag32 L.17. Please update the reference Rannik et al. (2016). The article is now published in AMT.

Caption of Table 2. Do you mean the std of lateral wind speed?

Figure 2 . Please explain in the caption what are those black arrows pointing down in fig2f.

Figure 6. Please reduced the scale of CH4 fluxes. It is very difficult to see how EC compare with ebullition data.

Please increase the font size in all figures.

References

Ono et al, 2008. Apparent downward CO2 flux observed with open-path eddy covariance over a non-vegetated surface.  Theor. Appl. Climatol. 92, 195–208

Peltola, O., Hensen, A., Helfter, C., Belelli Marchesini, L., Bosveld, F. C., van den Bulk, W. C. M., Elbers, J. A., Haapanala, S., Holst, J., Laurila, T., Lindroth, A., Nemitz, E., Röckmann, T., Vermeulen, A. T., and Mammarella, I.: Evaluating the performance of commonly used gas analysers for methane eddy covariance flux measurements: the InGOS inter-comparison field experiment, Biogeosciences, 11, 3163-3186, doi:10.5194/bg-11-3163-2014, 2014

---

## Referee Comment (RC2) · C. Wille (Referee) · 25 May 2017

General Comments

The manuscript presents a 2.5-year CO2 and CH4 flux data set from a fen and lake within a subarctic peatland ecosystem. GHG fluxes from aquatic ecosystems have been identified to contribute to a large but uncertain amount to the global GHG budget. Thus the presented study delivers important data and a substantial contribution to scientific progress within the scope of Biogeosciences. The scientific approach and the applied methods are valid, related scientific work is amply referenced, and the results and conclusions are presented in a well-structured way and in an appropriate style. I support the publication of the manuscript after minor revision.

Specific Comments

1. The manuscript suffers from occasional vagueness in definitions and nomenclature. For instance, "ice-out" is inconsistently used in the manuscript. On first use, in the abstract (p. 2, line 12), it denotes the point in time when the ice is completely thawed. Further into the manuscript "ice-out", "ice-out season", and "ice-out period" are used synonymously denoting the "thaw season" as defined in section 2.7. I suggest to stick to the nomenclature introduced and to replace "ice-out" and its variations with "thaw period" wherever this is applicable throughout the manuscript. Further, in section 2.7 it is not made very clear that the defined "thaw season" not only comprises the actual thawing of the lake ice, but also - and importantly - the subsequent initial overturning of the lake water.

2. Gap filling of highly variable CH4 fluxes is notoriously challenging but at the same time crucial for determining seasonal and annual balances. Therefore, the artificial neural network (ANN) gap filling method should be well presented and discussed. I consent to the detailed description of the ANN method being given in the supplement so as to keep the text concise. However, the environmental drivers used for the ANN is important information which should be included in the main manuscript. Hence I suggest to move Table S2 to Appendix B (which in consequence could be reduced in text). Furthermore, a paragraph on the ANN gap filling performance should be added to the results section, and Fig. B1 should be part of it (and should be improved for better legibility). Especially in case of CH4 fluxes, which span 2 orders of magnitude, a discussion of how the ANN performs - both in case of the slowly varying background flux and episodic high emission events - would be very interesting.

3. There was no gap-filling performed for lake CO2 fluxes after May 2013 due to low data coverage. However, according to table S1, CO2 flux data coverage was back to normal in 2014. Gap-filling should be resumed for 2014 data if at all possible.

4. I like the statistics of measured fluxes presented in figure 4. However, with a data

coverage of typically 30 %, how reliable is this information, i.e. how does the picture change when you look at the statistics of gap filled fluxes? The mean and median fluxes presented in section 3.2 may have to be interpreted with a certain caution, especially concerning the transient lake fluxes.

5. I think there is more potential in the ebullition flux data from bubble traps than the qualitative comparison presented in figure 6. I would like to see the attempt of a qualitative analysis in order to derive an estimate, of how much of the EC flux stems from diffusion and ebullition. Further, the thaw of the lake ice and the initial overturning of the lake water after the "ice-cover" period seem to be well separated in time. Hence I suggest to divide the spring emission peak into a portion which originates from the escape of gas bubbles trapped in the ice, and a portion which originates from the initial overturning of the lake water. This could help to explain the large differences between the total thaw season $CH_4$ emissions in 2013 and 2014.

6. How was eddy covariance raw data logged (type of data logger), and what exactly was done with $CH_4$ raw data during August 2013 - December 2014? As I understand, $CH_4$ concentration was taken from FGGA raw data files and had to be synchronized and combined with sonic anemometer data before being fed into EddyPro. If this is the case, have you checked if this caused any bias in the flux calculation? Please clarify.

7. In addition to the maintenance-caused gap during February - March 2014, there is a large gap in $CH_4$ flux data during December 2013 - February 2014. Was all data of this period rejected by the quality screening? The same question arises for $CO_2$ fluxes during February - March 2014. Please clarify.

8. The "Burba effect" seriously compromises cold season $CO_2$ flux data from the LI-COR Li-7500 which you used. The fact that the "Burba correction" was not applied is important information and should be given in the methods section and not as a sideline in the discussion. To my knowledge, many researchers failed to derive a meaningful flux correction using Burba's method, in which case there is no other way than to use

the CO2 flux data as it is. However, since you used a Los Gatos FGGA analyzer, you could use its CO2 data to calculate another CO2 flux data set to use during winter or to confirm winter time fluxes from your Li-7500. Has this been attempted?

9. The manuscript would benefit from focusing and shortening. Some examples are given in the next section.

Technical Corrections

p. 3, line 8: Change "explains" to "explain".

p. 3, line 8: Should read "order-of-magnitude-scale uncertainty"; consider simplifying to "large uncertainty".

p. 3, line 30: Change "lake" to "lakes".

p. 5, line 3: Change "lake" to "lakes".

p. 5, line 27: "May be" sounds very weak. The cited paper must have a stronger opinion on this matter?

p. 6, line 4: Change "palsa" to "palsas".

p. 6, lines 4-5: Change the order to "During snow melt, there is a small surface inflow feeding...".

p. 6, line 19: Add "height" after "2.50 m".

p. 8, line 11: Remove "the" between "footprint" and "model".

p. 9: The first paragraph and the last sentence of section 2.5 could be deleted.

p. 9, lines 21-22: "The goodness of fit was quantified with ... the absolute root mean square error (RMSE)." "Absolute" is superfluous and can be deleted. But in fact, table S2 gives the RMSE in %, and it is unclear what these percentages refer to. I strongly recommend to give the RMSE in flux units.

p. 9, line26: Replace "per" by "with".

p. 10: Section 2.6 could be shortened drastically by focusing on the reliability of the low (winter time) fluxes and on a brief outline of the error propagation method and the bias error.

p. 11, lines 21-22: Simplify "daily energy input (upwelling > downwelling radiation)" to "mean daily net radiation".

p. 11, line 22: Wrong reference. Change Fig. 2b to Fig. 2c

p. 11, line 23: Wrong reference. Change Fig. 2c to Fig. 2d

p. 11, line 23: Change albedo from 5 % to 0.05 to be consistent with units in Fig. 2.

p. 12, lines 9-10: Differences in mean temperature correlate with differences in total net radiation, or more simply, mean temperatures correlate with total net radiation values. Please correct.

p. 12, line 14: "thermal stratification along lake depth" sounds odd. Consider changing to "thermal stratification of the lake". (Again in line 16)

p. 12, line 15: "... large" thermal stratification ..." Consider replacing "large" by "strong" if that is what you mean.

p. 12, lines 14-15: Replace "was repeated each year" by "was similar in both years".

p. 12, lines 15, 16, 22: Wrong reference. Change Fig. 2d to Fig. 2f.

p. 13, line 2: Replace "followed" by "showed".

p. 13, line 12: Delete "but".

p. 13, lines 13-14: "The highest $CO_2$ uptake rates were observed during the summer of 2014, which was the warmest summer of the study period with highest solar radiation input (Table 1)." Table 1 lists only total net radiation values. As solar radiation can be expected to have a much higher explaining power for carbon fluxes (as confirmed by

its inclusion in the correlation analysis, table 2), total solar radiation should be reported in table 1.

p. 14, line 15: The correlation between increases in sediment temperature and CH4 bursts from the lake can hardly be seen – I suggest to delete this sentence. The correlation with falling atmospheric pressure described in line 7 is much better visible.

p. 15, line 29: Replace "lead" by "led".

p. 16, line 1: I would not expect a complete ice cover at a fen dominated by vascular plants as described in the study site section. Unless the water table is very high at the onset of freezing. I suggest to rephrase this passage.

p. 16, lines 27-31: The passage on the Burba correction is pointless, because - as written at the end of the paragraph - it corrects fluxes towards higher values and so cannot explain the too high fluxes during the winter 2013-2014.

p. 17, lines 24-25: Mind the causal connection between temperature increase and decrease of CH4 solubility! Rephrase, e.g. "...since a seasonal increase in sediment temperature favors methanogenesis and additionally causes a decrease of CH4 solubility..."

p. 18, lines 11-20: The whole paragraph seems inconclusive – how does it relate to your data?

p. 19, line 2: Correct "release" to "released".

p. 19, line 23: Correct "term" to "terms".

p. 21, line 21: Change word order to "Alaskan thermokarst lakes".

p. 22, line 13: Change "period" to "periods".

Table 1: I suggest to move the dates from the figure caption to the table. Total solar radiation should be added as this is the most important driver of $CO_2$ fluxes during

ice-free periods (in which case total net radiation could be omitted). Tables 1 and 3 should be merged into one table.

Table 2, caption: Wrong reference. Change "Table 2" to "Table 1".

Figure 2, caption: Add "daily means of" where applicable. Explain shaded area, "PN" and arrows.

Figure 3: Add grid lines, or at least y=0 lines. This helps the reader to determine if small fluxes are positive or negative or fluctuate around zero.

Figs. 4, 6, 8, 9, B1: The axis labels are too small.

Figure 6: Remove temperature plots. The suggested correlation between sediment temperature and CH4 flux can hardly be seen anyway.

Figure 7: One of the two graphs can be omitted, as they show the same data.

Figure 8: I suggest to remove the data of single years. The great variability makes it difficult to extract the important information from the graphs. Add grid lines, or at least y=0 lines.

Figure C1: This figure could be deleted. There is no real gain of information compared to figure 4.

Table S2: RMSE is given in % - of what? Please use flux units. What is the mean random error given in the last table row? Please explain.

––––––––––––––––––––

---

## Author Comment (AC1) · 23 Jun 2017

On behalf of all authors, I thank the two referees for a thorough evaluation of the manuscript with relevant and constructive comments. Following are specific responses to each of the referees' comments.

*The referees' comments are in black and the author's responses in red*

**Response to referee comment #1**

This manuscript presents two years of methane and carbon dioxide flux measurements at a lake and a fen sites. The dataset is very interesting, the framework analysis and results discussion very comprehensive and well written. I can recommend the final publication in Biogeosciences after the following comments are properly addressed:

1) Measurements of $CO_2$ fluxes were done by using an open path IRGA (LI-7500). Although this is a convenient or sometimes the only option in remote sites, I would use it with caution for measuring very low fluxes like the one presented here for the lake. The effect of air density fluctuations on $CO_2$ flux becomes very important, and the validity of WPL correction for low $CO_2$ flux has been also questioned in the past (see Ono et al., 2008). The authors are aware of this and that is why efforts were made to quantify the associated uncertainties on individual fluxes but also on annual sums, and why a strict approach was chosen when filtering the flux dataset. The dataset has been thoroughly quality checked based on micrometeorological and statistical criteria, as much as it was possible with the available data and instrumentation.

2) I was a bit surprised to see that the high frequency corrections for $CO_2$ flux were so high (31% on average). Usually for open path EC system is much less. I guess this may be because of relatively large separation between the IRGA and the sonic anemometer. What was the separation? What is the value range of time lag for $CO_2$ and which windows have you used? There may be problems with the WPL temperature term, which in theory should be measured in (or close) the path of the IRGA? This is a valid point, indeed the separation between the two instruments is large (42 cm vertical separation, 26 cm northward separation and 35 cm eastward separation). The nominal time lag between vertical wind speed and $CO_2$ was set to 0s, with a searching window from -4s to 4s. There was no thermistor close to the IRGA, thus the ambient temperature measured at a nearby mast was used for WPL correction and throughout the paper. The effect of the WPL correction on the $CO_2$ fluxes is quantified in Appendix A.

3) What would be the reason of relatively high (anti)-correlation between $CO_2$ flux and H during wintertime (from Table 2)? And what about summer? A possible explanation for the anti-correlation between $CO_2$ flux and H during summer is discussed in the last paragraph of section 4.2.1. We suggested that it could be due to the diffusive $CO_2$ flux (in this case, downward) between the surface and the atmosphere being enhanced by waterside convection (denoted by positive H), as it has been shown in other lake studies where it was associated with an evasion of $CO_2$. The correlation in winter, however, remains to this date unexplained, and may be linked to instrumental issues.

4) $CH_4$ flux: it is not clear if $H_2O$ was measured by LGR. If not, then I guess the $H_2O$ fluxes

measured by LI-7500 were used in the WPL correction for $CH_4$. What EddyPro does when $H_2O$ comes from the LI-7500? The $H_2O$ fluctuations in the sampling cell of LGR may be quite different than the ones measured by the open path IRGA. How the authors cope with this issue? Why the compensation for the pressure term was also added? Which pressure data have been used for this? Thank you for insisting on this point, which led us to find an overlook in the flux calculation. In EddyPro version 5.2, a revision of the WPL formulation by Ibrom et al. (2007) is proposed for closed path instruments. The pressure term is an addition by EddyPro to the original formulation. Air pressure measured at a nearby mast was input to the software for this purpose.

We unfortunately did not use $H_2O$ data from the FGGA in this study due to a faulty electronic connection at the time of data collection, thus we relied on the open path LI7500 analyser for $H_2O$ measurements. EddyPro requires a metadata file with information on instrument model, inlet tube properties, sensor separation, etc. We therefore assumed the open-source software to proceed and correct accordingly by taking these metadata information into account. However, after further investigation motivated by the referee's enquiry, we realized that EddyPro does not seem to apply the WPL correction when $H_2O$ measurements from the same closed path instrument are not available. This was a surprise considering what the text in the manual of the software version 5.2 implies (i.e. the application of a classic *a posteriori* WPL approach if cell data are lacking for closed path instruments). It actually appears, after recent verification, that in EddyPro a lack of available $H_2O$ data from the same closed path instrument results in no WPL correction. This is unfortunately not clear in the EddyPro settings, where the user chooses to apply the correction or not, and to this date we are unsure on how the matter is handled in version 5.2.

We thus performed a test on summer $CH_4$ flux data to evaluate the magnitude of the density effects on $CH_4$ fluxes during the summer of 2014. We were able to recover part of the $H_2O$ data from the FGGA using the same synchronization method as for $CH_4$ (cf. methods section of the paper) and used these measurements to apply the WPL correction in EddyPro using the formulation by Ibrom et al. (2007). The result is a difference of about 1% in flux magnitude - see also Figure A of this document.

The low magnitude of the WPL correction can be expected here, due to the long sampling line that attenuates significantly the $H_2O$ signal as well as temperature and pressure fluctuations thus density effects. Hence, the correction would have likely a minimal impact on the $CH_4$ flux dataset.

In the revised version of the manuscript, Appendix A will be corrected to acknowledge this overlook and a quantification of the estimated (minimal) effect of density fluctuations on closed path $CH_4$ flux data will be added.

[Figure]

Figure A: Comparison of $CH_4$ fluxes with and without WPL correction for the year 2014.

5) Besides of density fluctuations caused by $H_2O$, spectroscopic correction should be also applied to $CH_4$ flux (see Peltola et al., 2014). It seems to be some correlation between $CH_4$ flux and LE (table 2). Is this because of points 4 and 5? The correlation between $CH_4$ flux and LE is only notable during the ice-free season. It would be expected to be consistent over the year if it was due to a systematic error or bias present in the full dataset. We were not aware of the spectroscopic effect as an additional correction applied to $CH_4$ fluxes measured with closed path systems. Again, EddyPro offers this correction for open path $CH_4$ analyzers only. That said, having to use open-path $H_2O$ data may introduce more uncertainty than without the application of the spectroscopic effect.

Summarizing, I know that there are limitations on including (in proper way) all these aspects, but at least it should be checked how important they are. Finally, the authors should acknowledge more clearly these aspects when discussing the uncertainty of these fluxes. We agree that those are important technical points and appreciate that the referee emphasizes it. An attempt is made in our study to quantify the importance of the density effects and spectral corrections (cf. Appendix A). In the revised version of the manuscript, these points and their potential effects on the fluxes will be made clearer. The flux dataset has been thoroughly checked and, while acknowledging uncertainties, we remain confident in the $CH_4$ flux dataset due to strong agreement with other data sources (chambers at the fen and ebullition traps at the lake); as for $CO_2$ fluxes, we provided the best estimates possible with the available instrumentation at the time of the study. We will emphasize more clearly the associated uncertainties in the revised manuscript.

Minor comments:

Pag 6 L 11. Add also latent heat flux. OK

Pag6 L.18. "mixing ratio" usually means "dry mole fraction", but I guess this is "wet mole fraction", so without dilution correction. For calculating dry mole fraction point by point (high frequency data), simultaneous $H_2O$ measurements are needed. This is correct and will be corrected

in the text accordingly.

Pag. L.26-28. How the synchronization was done? Just using the timestamps? Raw data stored on the FGGA memory are not sampled at exactly 10Hz but at a variable frequency (11 to 12Hz). Raw $CH_4$ data stored on the FGGA were thus linearly interpolated on 10Hz timestamps. Additionally, to prevent mistakes due to a potentially uncalibrated clock on the FGGA, we did not use only the timestamp to synchronize the dataset. It was done in half-hour moving chunks of data by maximizing the correlation between logger data and FGGA data. The time showing the best correlation was chosen as a reference to adjust the clock, then $CH_4$ data were linearly interpolating onto the correct (logger) timestamp. Thus, when computing fluxes, the time lag for $CH_4$ fluxes between September 2013 and December 2014 was set to be searched within a large window that included 0s. The synchronization procedure was quality-checked after flux computation, cf. response to referee #2, point 6.

Pag.7 L.14. FST<0.3 is quite strict criteria. What about using FST<1? Is there a relevant difference in the data coverage? A strict filtering approach was chosen due to challenging footprint conditions, to ensure that fluxes actually represented the surface of interest. Adopting FST<0.3 as a criteria resulted in removal of outliers and negative values.

Pag.7 L.15. Ustar threshold is taken equal to 0.1 m/s. How this was determined? The threshold was determined using the online tool available at https://www.bgc-jena.mpg.de. This precision together with a reference will be added in the text.

Pag.8 L8-10. Could you report some of these values used for the footprint calculation, e.g. roughness lengths? A dynamic roughness length was used to represent the evolution of the canopy height (and presence of snow) on the fen side throughout the year, while a constant roughness length was adopted for the lake side. The values will be added in the text.

Pag.8 L13-14. Is longer footprint in winter because of more stable conditions (negative H and low wind speed)? Or why? Longer footprint in winter is most probably due to a lower roughness length (snow cover), which is a user-defined parameter, but also to more stable conditions, since H is negative during most of the winter. Indeed, the stability parameter is higher in winter increased (average of 0.10) in comparison to the summer season (average of 0.02). A sentence will be added in the text.

Pag.8 L16. Based on what, the criterium σv < 1 m/s was used? This criterium was used to limit lateral contamination of $CO_2$ fluxes into the footprint area of interest. The threshold was used after Forbrich et al (2011), who used this criterium to remove high crosswind fluctuations in their footprint study. The reference will be added.

Pag.9L.25. Why the method did not perform well for lake $CO_2$ flux? As shown by the density distribution in Figure C1 of the paper, lake $CO_2$ fluxes were very low and close to zero most of the time. Furthermore, they comprised a large amount of gaps, partly due to strict data filtering. We therefore decided to exclude the ANN results for the lake $CO_2$ fluxes as we found them to introduce a very high and unnecessary uncertainty.

Pag10L12-13. The random error of fluxes is usually proportional to the flux magnitude. Do you mean the relative random error (error normalized by the flux) is smaller? Yes, correct. It is proportional to the magnitude of the flux but relatively to the flux of smaller importance when fluxes are higher. This will be corrected in the new manuscript.

Pag.10 L21. How did you calculate the RE of the fluxes modeled with ANN? Each value used for gap filling is the mean of several ANN model runs (cf. Text S1). The 25 best runs (according to $r^2$) were averaged to output the modeled fluxes used in the gap filling. The standard deviation of these 25 model outputs was used as a quantification of the random error of each value used for gap filling. The average of these individual random errors was then computed as the mean random error for each modeled series ($CH_4$ fen, $CH_4$ lake, $CO_2$ fen). Information will be added to Text S1 and Table S1 in the revised manuscript.

Pag.11. L9. Is the thaw season the same as ice-out season? Yes. The term will be replaced throughout the manuscript for coherence, using "thaw season" as defined in section 2.7.

Pag.12 L15-16. Do you mean Fig. 2e and Fig 2f ? Yes. This will be corrected.

Pag. 13. L15. How do you explain this fall burst of $CO_2$? The warmer summer in 2014 may have caused a thermal stratification at the end of the season not present in other years. This could result in an accumulation of $CO_2$ and a subsequent degassing when lake cooling in fall triggers water mixing.

Pag.15 L5-10. The diel cycle of 2012 H is quite noised respect to the other years. Why? H flux data coverage in the months of June-July-August was 5% lower in 2012 as compared to 2013 and 2014, thus the seasonal diel cycle may be more sensitive to variability between days.

Pag.16 L10-20. May be some literature values can be added here for comparison. Values will be added.

Pag.16 L22-23. The highest correlation I can see for winter fluxes is with H. This is true, but it does not invalidate the observation made in this sentence. The correlation in winter with H is yet unexplained and could potentially be instrument-related but we have no mean of quantifying it.

Pag.16 L24. With EC it is not possible to measure advection, however You may see an increase of $CO_2$ mean concentration in the data, which may indicate non-turbulent transport of $CO_2$ from land. Thank you for the suggestion. After verification, there is an increase in $CO_2$ concentration in winter, as compared to summer values. This will be commented in the text.

Pag.16 L.27-31. I would say that it could be important to get a rough estimate of this correction. The effect (and direction) of this correction depends on the sign of H. We will make a rough estimate of the correction and discuss it.

Pag.17 L.9-11. How the EBC (energy balance closure) plots look like in different years? EBC was not computed for the lake side because of the uncertainty related to the computation of heat storage in the lake with the available data. As a hint on the eddy covariance system performance, Figure B

below shows the energy balance closure on the fen side on half-hourly and daily time scales (full dataset). If the allocated time to revise the manuscript allows it, we will attempt to compute an ECB on the lake side too.

[Figure]

Figure B: Energy balance closure on the fen side at half-hourly (a) and daily (b) time scales.

Pag.20 chapter 4.3.2. Are there any measurements of $pCO_2$ for this lake during summer? Or chamber measurements? Anything that could support the sink that you measured with EC? Unfortunately, there were no coincident measurements of $pCO_2$ in the lake water during the study period. A new study is currently measuring $pCO_2$ along with chamber and EC measurements, which will be able in the future to validate or not the $CO_2$ sink.

Pag.23 L17. Coordinate rotation is not really a correction. True, this will be corrected.

Pag.23 L19-22. Please report the values of time windows and time lag. Was the lag maximization applied also for sonic temperature? The lag between w and Ts should be 0. The lag maximization was not applied for sonic temperature (time lag set at 0s). The sentence will be modified for clarity and time lag windows reported.

Pag32 L.17. Please update the reference Rannik et al. (2016). The article is now published in AMT. This will be corrected.

Caption of Table 2. Do you mean the std of lateral wind speed? Yes; this will be corrected.

Figure 2 . Please explain in the caption what are those black arrows pointing down in fig2f. Black arrows indicate the estimated time of full overturn. This will be added in the caption.

Figure 6. Please reduced the scale of $CH_4$ fluxes. It is very difficult to see how EC compared with ebullition data. The figure will be redrawn to improve visibility.

Please increase the font size in all figures. Font size will be increased where needed.

**Response to referee comment #2**

General Comments

The manuscript presents a 2.5-year $CO_2$ and $CH_4$ flux data set from a fen and lake within a subarctic peatland ecosystem. GHG fluxes from aquatic ecosystems have been identified to contribute to a large but uncertain amount to the global GHG budget.

Thus the presented study delivers important data and a substantial contribution to scientific progress within the scope of Biogeosciences. The scientific approach and the applied methods are valid, related scientific work is amply referenced, and the results and conclusions are presented in a well-structured way and in an appropriate style. I support the publication of the manuscript after minor revision.

Specific Comments

1. The manuscript suffers from occasional vagueness in definitions and nomenclature. For instance, "ice-out" is inconsistently used in the manuscript. On first use, in the abstract (p. 2, line 12), it denotes the point in time when the ice is completely thawed. Further into the manuscript "ice-out", "ice-out season", and "ice-out period" are used synonymously denoting the "thaw season" as defined in section 2.7. I suggest to stick to the nomenclature introduced and to replace "ice-out" and its variations with "thaw period" wherever this is applicable throughout the manuscript. Further, in section 2.7 it is not made very clear that the defined "thaw season" not only comprises the actual thawing of the lake ice, but also - and importantly - the subsequent initial overturning of the lake water. For clarity, the term "thaw period" will be chosen and coherently used throughout the manuscript.

2. Gap filling of highly variable $CH_4$ fluxes is notoriously challenging but at the same time crucial for determining seasonal and annual balances. Therefore, the artificial neural network (ANN) gap filling method should be well presented and discussed. I consent to the detailed description of the ANN method being given in the supplement so as to keep the text concise. However, the environmental drivers used for the ANN is important information which should be included in the main manuscript. Hence I suggest to move Table S2 to Appendix B (which in consequence could be reduced in text). Agreed, the table will be moved to Appendix B and the text adjusted accordingly. Furthermore, a paragraph on the ANN gap filling performance should be added to the results section, and Fig. B1 should be part of it (and should be improved for better legibility). Font size in figure B1 will be increased and the performance reported in Appendix B will be moved to a short paragraph in the results section.

Especially in case of $CH_4$ fluxes, which span 2 orders of magnitude, a discussion of how the ANN performs - both in case of the slowly varying background flux and episodic high emission events - would be very interesting. This is briefly discussed when emphasizing how the performance of the ANN method differed between the fen $CH_4$ flux dataset and the lake $CH_4$ flux dataset. We agree that ANN gap filling, as a rare method of gap filling EC fluxes and especially of lake EC fluxes (to our knowledge, we are the first to present an application of it to lake $CH_4$ fluxes), needs to be

appropriately described and discussed. We attempted to give enough information while keeping it concise, since the gap filling itself is not the primary focus of this paper. The method used closely followed the procedure introduced in details by Dengel et al. (2013).

3. There was no gap-filling performed for lake $CO_2$ fluxes after May 2013 due to low data coverage. However, according to table S1, $CO_2$ flux data coverage was back to normal in 2014. Gap-filling should be resumed for 2014 data if at all possible. Please note that $CO_2$ fluxes from the lake were not gap filled as the rest of the dataset using a gap filling model (see response to referee #1 for further detail), but by using mean fluxes per season. This can be considered valid for normally distributed dataset, but introduces a largest bias when data coverage is lowest. The second year (June 2013 - May 2014) comprises too many gaps. Depending on the allocated time for the revision of the manuscript, we will consider extending it to the last two seasons of the dataset when data coverage increased (spring and summer 2014), but we will not compute the second annual sum.

4. I like the statistics of measured fluxes presented in figure 4. However, with a data coverage of typically 30 %, how reliable is this information, i.e. how does the picture change when you look at the statistics of gap filled fluxes? From a statistical point of view, these results are valid, since eddy covariance produces a lot of data points, even with 30% data coverage over a year. I would argue that computing the correlation statistics on the gap filled datasets would not be appropriate and would give bias results, since the environmental parameters tested here are some of the same variables used to develop the gap filling models. This is why we chose to compute correlation statistics on measured fluxes only. These are a statistical exploration of the available dataset; generalizing it to the whole period is indeed uncertain, considering the data coverage, and is not necessarily the intention here.

The mean and median fluxes presented in section 3.2 may have to be interpreted with a certain caution, especially concerning the transient lake fluxes. Indeed, the mean alone would not be an accurate summary of the flux magnitude, since it is affected by the occasional large degassing in the lake dataset (cf. Figure C1). Thus, we reported medians along with the means, and underlined at various stages of the manuscript (and in Figure 4) the skewed distribution of the lake flux data.

5. I think there is more potential in the ebullition flux data from bubble traps than the qualitative comparison presented in figure 6. I would like to see the attempt of a qualitative analysis in order to derive an estimate, of how much of the EC flux stems from diffusion and ebullition. Data are not available at this level of details for the present manuscript. An ongoing study will quantify the part of diffusive and ebullition flux within the EC dataset by means of comparison with simultaneous chamber and bubble measurements. These results are presented in a separate study and so not included here. We hope the reviewer will accept that these studies are being kept separate.

Further, the thaw of the lake ice and the initial overturning of the lake water after the "ice-cover" period seem to be well separated in time. Hence I suggest to divide the spring emission peak into a portion which originates from the escape of gas bubbles trapped in the ice, and a portion which

originates from the initial overturning of the lake water. This could help to explain the large differences between the total thaw season $CH_4$ emissions in 2013 and 2014. This is a very good point that has actually been addressed in details in a previous study focusing specifically on the $CH_4$ degassing from the lake during the spring of 2013. In Jammet et al. (2015), we present a quantitative and qualitative analysis of the spring efflux in 2013 by suggesting a separation of the degassing event in three steps, which likely correspond to emissions from different gas sources (liberation of bubbles from the ice, diffusion from the water, overturn). A sentence in the discussion refers the reader to this paper for further information (section 4.2.2). Section 4.2.2 will be rephrased to make this point clearer.

6. How was eddy covariance raw data logged (type of data logger),

A CR1000 was used until June 2013. Occasional data loss occurred in 2012 due to the low performance of the CR1000 for heavily instrumented sites (skipping logging rows), which lead to its replacement with a CR3000 in June 2013. Data logger information will be added to the methods section.

and what exactly was done with $CH_4$ raw data during August 2013 - December 2014? As I understand, $CH_4$ concentration was taken from FGGA raw data files and had to be synchronized and combined with sonic anemometer data before being fed into EddyPro. If this is the case, have you checked if this caused any bias in the flux calculation? Please clarify. As explained in replied to referee #1 (see above), $CH_4$ raw data from the FGGA were synchronized with the high frequency data logged onto the data logger, then linearly interpolated to match the logger time stamp. After flux calculation, half-hours for which the synchronization procedure could not yield any reliable flux (i.e. where the method failed) were identified (lack of a significant peak in the cross-covariance function, as in Wienhold et al. (1994), Nordbo et al. (2012), Rinne et al. (2007)) and filtered out. This screening step was considered as a quality check of the synchronization procedure.

7. In addition to the maintenance-caused gap during February - March 2014, there is a large gap in $CH_4$ flux data during December 2013 - February 2014. Was all data of this period rejected by the quality screening? The same question arises for $CO_2$ fluxes during February - March 2014. Please clarify. Thank you for raising this point. The next version of the manuscript will include (in supplementary information) a further figure visualizing the data coverage for both instruments, indicating which time periods were flagged due to power outage or lack of data and which data were removed due to quality check procedures. During winter, measurements from the sonic anemometer and FGGA analyzer were regularly subject to multiple drop-outs and out of ranges values (filtered out during screening), most likely due to frost on the sonic and very cold air input to the analyzer, which does not perform well at low temperature.

8. The "Burba effect" seriously compromises cold season $CO_2$ flux data from the LI-COR Li-7500 which you used. The fact that the "Burba correction" was not applied is important information and should be given in the methods section and not as a sideline in the discussion. Agreed, this information will be moved to methods section. To my knowledge, many researchers failed to derive a meaningful flux correction using Burba's method, in which case there is no other way than to use

the $CO_2$ flux data as it is. However, since you used a Los Gatos FGGA analyzer, you could use its $CO_2$ data to calculate another $CO_2$ flux data set to use during winter or to confirm winter time fluxes from your Li-7500. Has this been attempted? Unfortunately, a problem with the data transfer on the $CO_2$ channel from the FGGA during the time of this study prohibited us to use the data for $CO_2$ flux computation.

9. The manuscript would benefit from focusing and shortening. Some examples are given in the next section.

Technical Corrections

p. 3, line 8: Change "explains" to "explain". OK

p. 3, line 8: Should read "order-of-magnitude-scale uncertainty"; consider simplifying to "large uncertainty". OK

p. 3, line 30: Change "lake" to "lakes". OK

p. 5, line 3: Change "lake" to "lakes". OK

p. 5, line 27: "May be" sounds very weak. The cited paper must have a stronger opinion on this matter? This will be rephrased to "According to Olefeldt and Roulet (2012), the two ecosystems are hydrologically connected…".

p. 6, line 4: Change "palsa" to "palsas". OK

p. 6, lines 4-5: Change the order to "During snow melt, there is a small surface inflow feeding...". OK

p. 6, line 19: Add "height" after "2.50 m". OK

p. 8, line 11: Remove "the" between "footprint" and "model". OK

p. 9: The first paragraph and the last sentence of section 2.5 could be deleted. The first paragraph will be shortened and the last sentence will be deleted.

p. 9, lines 21-22: "The goodness of fit was quantified with...the absolute root mean square error (RMSE)." "Absolute" is superfluous and can be deleted. But in fact, table S2 gives the RMSE in %, and it is unclear what these percentages refer to. I strongly recommend to give the RMSE in flux units. OK

p. 9, line26: Replace "per" by "with". OK

p. 10: Section 2.6 could be shortened drastically by focusing on the reliability of the low (winter time) fluxes and on a brief outline of the error propagation method and the bias error. The section will be shortened, to the extent that it keeps the information necessary to understand the results in e.g. Table 3.

p. 11, lines 21-22: Simplify "daily energy input (upwelling > downwelling radiation)" to "mean daily net radiation". OK

p. 11, line 22: Wrong reference. Change Fig. 2b to Fig. 2c OK

p. 11, line 23: Wrong reference. Change Fig. 2c to Fig. 2d OK

p. 11, line 23: Change albedo from 5 % to 0.05 to be consistent with units in Fig. 2. OK

p. 12, lines 9-10: Differences in mean temperature correlate with differences in total net radiation, or more simply, mean temperatures correlate with total net radiation values. Please correct. OK

p. 12, line 14: "thermal stratification along lake depth" sounds odd. Consider changing to "thermal stratification of the lake". (Again in line 16) OK

p. 12, line 15: "... large" thermal stratification ..." Consider replacing "large" by "strong" if that is what you mean. OK

p. 12, lines 14-15: Replace "was repeated each year" by "was similar in both years". OK

p. 12, lines 15, 16, 22: Wrong reference. Change Fig. 2d to Fig. 2f. OK

p. 13, line 2: Replace "followed" by "showed". OK

p. 13, line 12: Delete "but". OK

p. 13, lines 13-14: "The highest $CO_2$ uptake rates were observed during the summer of 2014, which was the warmest summer of the study period with highest solar radiation input (Table 1)." Table 1 lists only total net radiation values. As solar radiation can be expected to have a much higher explaining power for carbon fluxes (as confirmed by its inclusion in the correlation analysis, table 2), total solar radiation should be reported in table 1. Total solar radiation will be added to Table 1.

p. 14, line 15: The correlation between increases in sediment temperature and $CH_4$ bursts from the lake can hardly be seen − I suggest to delete this sentence. Although it is not systematic, a large part of the high $CH_4$ release events are preceded by an increase in surface sediment temperature and this is nice to show. The scale of $CH_4$ fluxes in this figure may be too large at the moment to properly identify this; it will be changed so that the variations in fluxes are clearer to the reader. The sentence will be rephrased. The correlation with falling atmospheric pressure described in line 7 is much better visible.

p. 15, line 29: Replace "lead" by "led". OK

p. 16, line 1: I would not expect a complete ice cover at a fen dominated by vascular plants as described in the study site section. Unless the water table is very high at the onset of freezing. I suggest to rephrase this passage. The water table is always high at this site; however, the ice cover is not complete, because stems and branches are sticking out of the snow. That is what the sentence meant and it will be rephrased for clarity.

p. 16, lines 27-31: The passage on the Burba correction is pointless, because – as written at the end of the paragraph - it corrects fluxes towards higher values and so cannot explain the too high fluxes during the winter 2013-2014. This sentence will be deleted and the mention of the self-heating effect will be moved to the methods section, as earlier suggested by the referee.

p. 17, lines 24-25: Mind the causal connection between temperature increase and decrease of $CH_4$ solubility! Rephrase, e.g. "since a seasonal increase in sediment temperature favors methanogenesis and additionally causes a decrease of $CH_4$ solubility..." OK

p. 18, lines 11-20: The whole paragraph seems inconclusive – how does it relate to your data? The paragraph is an attempt at explaining the (anti-)correlation observed between H and $CO_2$ flux during summer in our lake data. Additionally, it underlines how this relationship differs from what was shown in previous studies and suggest an explanation for this difference. The paragraph will be shortened and rewritten for clarity.

p. 19, line 2: Correct "release" to "released". OK

p. 19, line 23: Correct "term" to "terms". OK

p. 21, line 21: Change word order to "Alaskan thermokarst lakes". OK

p. 22, line 13: Change "period" to "periods". OK

Table 1: I suggest to move the dates from the figure caption to the table. Total solar radiation should be added as this is the most important driver of $CO_2$ fluxes during ice-free periods (in which case total net radiation could be omitted). Tables 1 and 3 should be merged into one table. Total net radiation will be replaced by total solar radiation. The dates may be added to Table 1 and Table 1 may be merged with Table 3 if a wide table on a landscape layout is acceptable for the editors of Biogeosciences.

Table 2, caption: Wrong reference. Change "Table 2" to "Table 1". OK

Figure 2, caption: Add "daily means of" where applicable. Explain shaded area, "PN" and arrows. OK

Figure 3: Add grid lines, or at least y=0 lines. This helps the reader to determine if

small fluxes are positive or negative or fluctuate around zero. Lines marking y = 0 will be added on Figure 3.

Figs. 4, 6, 8, 9, B1: The axis labels are too small. Font size will be increased where needed.

Figure 6: Remove temperature plots. The suggested correlation between sediment temperature and $CH_4$ flux can hardly be seen anyway. As mentioned earlier, Figure 6 will be modified to improve visibility. This will hopefully address this comment.

Figure 7: One of the two graphs can be omitted, as they show the same data. Figure 7b helps

visualizing the high correlation, which can be hard to see on Figure 7a due to the scale of the bars showing the spread of the data around the means. Figure 7b will be moved to supplementary information.

Figure 8: I suggest to remove the data of single years. The great variability makes it difficult to extract the important information from the graphs. Add grid lines, or at least y=0 lines. Single years are present to show that there is variability between years in terms of magnitude, yet the pattern is similar between years. To improve clarity, lines showing y = 0 will be added; single years will be suppressed from the figure and moved to supplementary information.

Figure C1: This figure could be deleted. There is no real gain of information compared to figure 4. Figure C1 is used to show the scale of the whole flux dataset and their distribution. In Figure 4, data are split between seasons. Figure C1 is referred to when commenting on the distribution of the dataset and to show the scale of the lake $CO_2$ fluxes in comparison with the other time series, which explains in part why we could not perform ANN gap filling.

Table S2: RMSE is given in % - of what? Please use flux units. OK What is the mean random error given in the last table row? Please explain. The mean random error given for each modeled series ($CH_4$ fen, $CH_4$ lake, $CO_2$ fen) is the mean of the standard deviation for each individual value used for gap filling. Please see our response to referee #1 for a more detailed explanation. The information will be added in the text and mentioned in the Table caption.

**References**

Dengel, S., Zona, D., Sachs, T., Aurela, M., Jammet, M., Parmentier, F. J. W., Oechel, W. and Vesala, T.: Testing the applicability of neural networks as a gap-filling method using $CH_4$ flux data from high latitude wetlands, Biogeosciences, 10(12), 8185–8200, doi:10.5194/bg-10-8185-2013, 2013.

Forbrich, I., Kutzbach, L., Wille, C., Becker, T., Wu, J. and Wilmking, M.: Cross-evaluation of measurements of peatland methane emissions on microform and ecosystem scales using high-resolution landcover classification and source weight modelling, Agric. For. Meteorol., 151(7), 864–874, doi:10.1016/j.agrformet.2011.02.006, 2011.

Ibrom, A., Dellwik, E., Larsen, S. E. and Pilegaard, K.: On the use of the Webb–Pearman–Leuning theory for closed-path eddy correlation measurements, Tellus B, 59(5), 937–946, doi:10.1111/j.1600-0889.2007.00311.x, 2007.

Jammet, M., Crill, P., Dengel, S. and Friborg, T.: Large methane emissions from a subarctic lake during spring thaw: Mechanisms and landscape significance, J. Geophys. Res. Biogeosciences, 120(11), 2015JG003137, doi:10.1002/2015JG003137, 2015.

Nordbo, A., Järvi, L. and Vesala, T.: Revised eddy covariance flux calculation methodologies – effect on urban energy balance, Tellus B, 64(1), 18184, doi:10.3402/tellusb.v64i0.18184, 2012.

Ono, K., Miyata, A. and Yamada, T.: Apparent downward $CO_2$ flux observed with open-path eddy

covariance over a non-vegetated surface, Theor. Appl. Climatol., 92(3–4), 195–208, doi:10.1007/s00704-007-0323-3, 2008.

Peltola, O., Mammarella, I., Haapanala, S. and Vesala, T.: Intercomparison of four methane gas analysers suitable for eddy covariance flux measurements, [online] Available from: http://www.nateko.lu.se/nordflux/pdf/denmark2011/Peltola_Olli_Nordflux_Nov2011.pdf (Accessed 24 September 2014), 2011.

Peltola, O., Hensen, A., Helfter, C., Belelli Marchesini, L., Bosveld, F. C., van den Bulk, W. C. M., Elbers, J. A., Haapanala, S., Holst, J., Laurila, T., Lindroth, A., Nemitz, E., Röckmann, T., Vermeulen, A. T. and Mammarella, I.: Evaluating the performance of commonly used gas analysers for methane eddy covariance flux measurements: the InGOS inter-comparison field experiment, Biogeosciences, 11(12), 3163–3186, doi:10.5194/bg-11-3163-2014, 2014.

Rinne, J., Taipale, R., Markkanen, T., Ruuskanen, T. M., Hellén, H., Kajos, M. K., Vesala, T. and Kulmala, M.: Hydrocarbon fluxes above a Scots pine forest canopy: measurements and modeling, Atmospheric Chem. Phys., 7(1), 3361–3372, doi:10.5194/acp-7-3361-2007, 2007.

Wienhold, F. G., Frahm, H. and Harris, G. W.: Measurements of N2O fluxes from fertilized grassland using a fast response tunable diode laser spectrometer, J. Geophys. Res., 99(D8), 16557–16567, doi:10.1029/93JD03279, 1994.